# Continual Parameter-Efficient Adaptation for Rehearsal-Free Graph Class-Incremental Learning

## Abstract

Graph Class-Incremental Learning (GCIL) seeks to learn novel classes sequentially while preserving knowledge acquired from previously seen classes. However, to tackle the pervasive challenge of catastrophic forgetting, recent GCIL methods often train separate classifiers from scratch for each task, which is redundant in design and computationally expensive. Moreover, isolating streaming data in different tasks hampers knowledge transfer across tasks. To address these dilemmas, we first propose Graph2Hyper, a parameter-efficient framework that utilizes a hypernetwork to generate task-specific classifiers on the fly based solely on the input graph of the current task. Concretely, the hypernetwork is composed of just two linear layers: a frozen, task-shared layer that preserves cross-task knowledge, and a trainable, task-specific layer that captures the unique characteristics of each task. To distinguish between different tasks in the incremental learning process, task-prototypes are extracted via pooling over global node representations, which capture task-specific contextual knowledge. To further model the association between tasks and corresponding classes, we construct class-prototypes with dynamic task-level bias through a learnable mapping function. By encoding class-level discrimination while retaining task-level context, the hypernetwork enables continual forget-free adaptation to new classes without the need for prototype rehearsal. Extensive experiments on four benchmark datasets demonstrate that Graph2Hyper achieves promising performance with superior parameter efficiency.

## 1 Introduction

Graph continual learning (GCL), also referred to as Graph Incremental Learning, has recently emerged as an important paradigm to extend graph neural networks (GNNs) toward dynamic and open-world scenarios (Zhang et al., 2024; Tian et al., 2024; Febrinanto et al., 2023; Zhang et al., 2022a). The goal is to continually learn a model that not only adapts to new and emerging graph data but also preserves the knowledge acquired from previous graph tasks, where each graph task consists of nodes belonging to a unique set of classes within a graph. Depending on how new knowledge is introduced, GCL can be categorized into two settings: *class-incremental learning* (GCIL), where new classes arrive sequentially across tasks, and *task-incremental learning* (GTIL), where task identities are explicitly available during testing. Compared to GTIL, the absence of task IDs in GCIL introduces an additional challenge, thereby leading to a significant performance gap between GCIL and GTIL.

Due to privacy concerns and the hardware limitations in storage and computation, GIL assumes that data from previous graph tasks is not accessible when learning new tasks. This restriction results in *catastrophic forgetting*, *i.e.*, degraded classification accuracy on previous tasks caused by model updates on new tasks. Existing approaches typically mitigate catastrophic forgetting by preserving the important parameters of previous tasks (Liu et al., 2021), continually expanding model parameters for new tasks (Zhang et al., 2022b; 2023b), or augmenting with a memory module for data replay (Zhou & Cao, 2021; Zhang et al., 2022c; 2023c; Niu et al., 2024a; Liu et al., 2023; Niu et al., 2024a). However, these approaches exhibit limited ability in handling intra-task class separation, resulting in unsatisfactory GCIL classification accuracy, especially on earlier tasks. Consequently, most existing GCL methods suffer substantially higher forgetting in GCIL than in GTIL.

Beyond these, TPP (Niu et al., 2024b) simplifies GCIL into a GTIL problem, achieving nearly 100% task ID prediction accuracy and a 0% forgetting ratio. However, methods like TPP require training separate classifiers from scratch for each task, which introduces considerable training and storage overhead. Moreover, given that a sequence of graphs often contains task-shared knowledge, existing approaches fail to explicitly leverage cross-task knowledge transfer in the architectural design.

In this paper, we propose **Graph2Hyper**, a parameter-efficient framework that employs a lightweight hypernetwork to dynamically generate task-specific classifiers conditioned on the input graph of each task. Concretely, to achieve coarse-to-fine inter-task separation and intra-task classification, Graph2Hyper first leverages task-oriented distribution matching to construct task-prototypes and further introduces class-prototypes with dynamic bias to enhance intra-task discrimination. To capture both task-shared commonalities and task-specific distinctions across a sequence of tasks, we adopt a hypernetwork to dynamically generate classifier parameters on the fly, while updating only a portion of the hypernetwork for each new task. We theoretically show that parameter sharing in the hypernetwork is equivalent to knowledge transfer across classification heads, and that such hypernetwork-based knowledge transfer preserves a tighter error upper bound in continual learning. Following the TPP setting, Graph2Hyper achieves a forget-free property while reducing trainable parameters. Extensive experiments on four benchmark datasets demonstrate the effectiveness of Graph2Hyper over state-of-the-art (SOTA) baselines. The contributions of this work are as follows:

- We propose a parameter-efficient framework for rehearsal-free GCIL, termed Graph2Hyper, which employs a lightweight hypernetwork-based adaptation with task-shared knowledge.

- We introduce both task- and class-prototypes for coarse-to-fine classification, and derive a tighter error upper bound with cross-task knowledge transfer.

- Extensive experiments validate the effectiveness of Graph2Hyper, showing superior performance compared to existing SOTA methods.

## 2    RELATED WORK

**Graph Continual Learning.** Graph Continual Learning (GCL) aims to integrate new knowledge into Graph Neural Networks (GNNs) while mitigating the problem of catastrophic forgetting, that is, the loss of previously learned information. Recent studies on GCL (He et al., 2023; Zhou & Cao, 2021; Liu et al., 2021; Sun et al., 2023; Wang et al., 2022; Su et al., 2023; Rakaraddi et al., 2022; Zhang et al., 2023b; Perini et al., 2022; Zhang et al., 2022c; 2023c;a) generally fall into three categories: regularization-based, parameter isolation-based, and replay-based methods. Regularization-based approaches constrain the learning process of new tasks by penalizing changes to parameters that are important for previous tasks. For instance, TWP (Liu et al., 2021) incorporates regularization terms to preserve key parameters in both topological aggregation and loss optimization from earlier tasks. Parameter isolation-based approaches (Zhang et al., 2023b), on the other hand, allocate separate sets of parameters to different tasks in order to avoid interference. In contrast, replay-based methods (Zhou & Cao, 2021; Zhang et al., 2022c; 2023c; Liu et al., 2023; Niu et al., 2024a) rely on an external memory buffer that stores task-related information and reuses it during training on new tasks. In our work, we propose a task-oriented distribution matching to construct task-prototypes and incorporate class-prototypes with dynamic bias adjustment to strengthen intra-task discrimination.

**Hypernetworks.** Hypernetworks (Ha et al., 2017) are neural networks designed to generate the parameters of a main deep neural network (DNN) instead of learning them directly. This paradigm has been widely explored in recent studies. For example, they have been applied to cold-start recommendation (Lin et al., 2021), image classification Przewięźlikowski et al. (2024), and graph neural networks (Brockschmidt, 2020), where hypernetworks modulate aggregation weights in message passing. Beyond single-task settings, hypernetworks have shown advantages in multi-task and transfer learning by enabling soft weight sharing across tasks. They also support data-adaptive modeling, tailoring the main network's weights to specific inputs, while requiring fewer parameters than conventional DNNs, which improves efficiency in resource-constrained scenarios. In this paper, we adopt hypernetworks for parameter-efficient adaptation, generating task-specific classifiers with substantially fewer trainable parameters compared to training all classifiers from scratch.

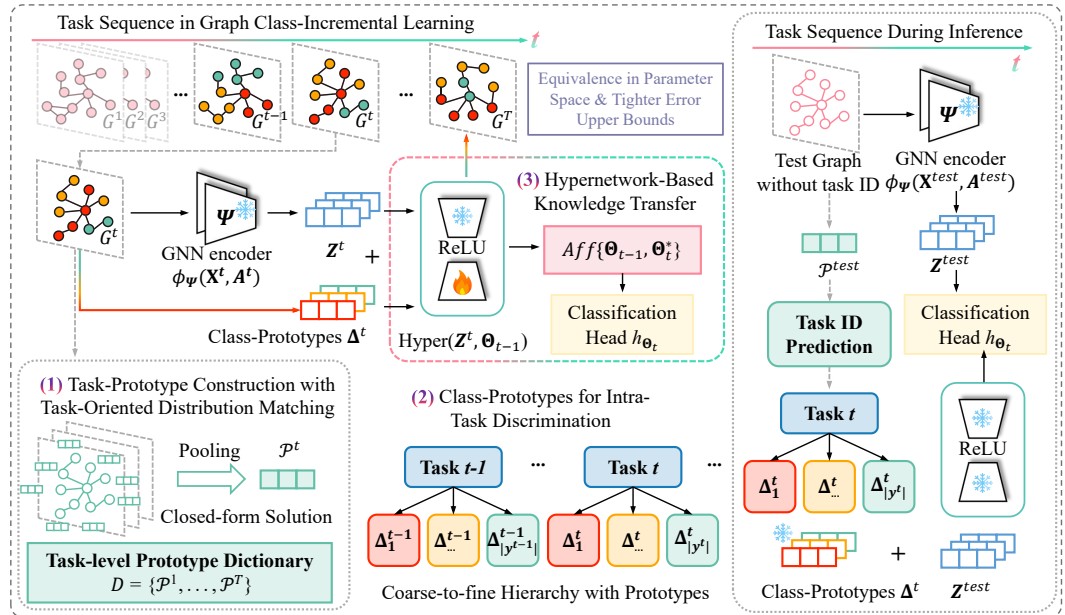

Figure 1: The framework of Graph2Hyper. During training, given a task graph $G^t$, the pre-trained GNN encoder extracts node representations $\mathbf{Z}^t$. (1) Task-oriented distribution matching constructs a shared task-prototype stored in a dictionary $D$. (2) Class prototypes $\Delta^t$ with dynamic biases refine node representations to improve intra-task discrimination. (3) A shared-parameter hypernetwork generates task-specific classifier heads while transferring knowledge across tasks, yielding a tighter generalization bound. During inference, the task-prototype of test graph $G^{\text{test}}$ is matched with dictionary prototypes to determine its task ID. The hypernetwork then generates the classifier parameters for node classification on $G^{\text{test}}$.

## 3 PRELIMINARIES

**GCIL Problem Formulation.** We consider a class-incremental learning scenario consisting of a sequence of tasks $\{G^1, \ldots, G^T\}$ where $T$ is the number of tasks. Each task $t$ is associated with a graph snapshot $G^t = (\mathbf{A}^t, \mathbf{X}^t)$, where $\mathbf{A}^t$ denotes the adjacency matrix, $\mathbf{X}^t \in \mathbb{R}^{N_t \times d}$ is the node attribute matrix with feature dimension $d$, and $y^t$ is the label set of nodes in $G^t$. Following the GCIL setting, each task introduces a disjoint set of classes, *i.e.*, $y^t \cap y^j = \varnothing, \forall t \neq j$, and hence the number of classes grows as $t$ increases. Since the task ID $t$ is unknown at inference time, the model must classify each test node to one of the accumulated $\sum_{j=1}^{T} |y^j|$ classes. The key challenge is to continually adapt to new tasks while avoiding catastrophic forgetting of prior knowledge.

**Pre-training for GNN Encoder.** Following the setup in TPP (Niu et al., 2024b), we adopt an unsupervised pre-trained GNN as the encoder and keep it frozen for subsequent tasks. Specifically, from the initial graph $G^1$, we construct training triples $(u, v_+, v_-)$, where $v_+$ is a positive node sampled from the local neighborhood of $u$, and $v_-$ is a negative node sampled from non-neighbors. The node similarity is estimated by a discriminator $g = \mathtt{MLP}(\mathrm{sim}(\cdot, \cdot))$. The pre-training goal is to increase the semantic similarity between the node embedding $\mathbf{z}_u$ and $\mathbf{z}_{v_+}$ while decreasing that between $\mathbf{z}_u$ and $\mathbf{z}_{v_-}$. We formalize the objective as,

$$\mathcal{L}_{\text{pre}}(\mathbf{\Psi}; \mathbf{Z}) = -\sum_{(u,v_+,v_-)} \ln \frac{\exp(g(\mathbf{z}_u, \mathbf{z}_{v_+})/\tau)}{\sum_{v \in \{v_+, v_-\}} \exp(g(\mathbf{z}_u, \mathbf{z}_v)/\tau)}, \tag{1}$$

where $\tau$ is the temperature, $\mathbf{Z} = \mathtt{stack}[\phi_{\mathbf{\Psi}}(\mathbf{X}_i, \mathbf{A}_i)]_{i=1}^{N_1}$ is the embedding obtained by the encoder.

## 4 METHODOLOGY

In this section, we elaborate on the proposed parameter-efficient framework Graph2Hyper, as shown in Figure 1. Graph2Hyper first leverages task-oriented distribution matching to construct task-prototypes,

and further introduces class-prototypes with dynamic bias to enhance intra-task discrimination. To enable continual adaptation on task-specific classifiers, a lightweight hypernetwork is employed to generate parameters with task-shared knowledge transfer, which allows Graph2Hyper to achieve a tighter generalization error bound.

## 4.1 HIERARCHICAL CLASSIFICATION WITH TASK- AND CLASS-PROTOTYPES

A natural solution to the GCIL problem is to first distinguish samples from different tasks and then further differentiate all classes within a given task. To this end, we sequentially construct task-prototypes and class-prototypes, and fully exploit their hierarchical relationships.

**Task-Prototype Construction with Task-Oriented Distribution Matching.** Given an input graph $G^{(t)} = \{\mathbf{A}, \mathbf{X}\}$, the non-parametric feature propagation module is deployed to smooth the node features with $K$-th order as $\mathbf{H} = \hat{\mathbf{A}}^K \mathbf{X}$, where $\hat{\mathbf{A}} = \tilde{\mathbf{D}}^{\frac{1}{2}} \tilde{\mathbf{A}} \tilde{\mathbf{D}}^{\frac{1}{2}}$ is the normalized adjacency matrix with self-loops, and $\tilde{\mathbf{D}}$ is degree matrix for $\tilde{\mathbf{A}} = \mathbf{A} + \mathbf{I}$. This propagation scheme can be replaced with other alternatives with diverse characteristics, such as SGC (Wu et al., 2019).

**Proposition 4.1.** After multi-hop propagation, let $\mathbf{Z} = \mathbf{H}\boldsymbol{\Psi} = \hat{\mathbf{A}}^K \mathbf{X}\boldsymbol{\Psi}$ and $\mathbf{Z}' = \mathbf{H}'\boldsymbol{\Psi}$ be the representations of the original graph $G^{(t)}$ and its task-prototype, respectively, and $\hat{\mathbf{P}}$ is the aggregation matrix. The distribution matching objective can be upper-bounded by:

$$\underset{\boldsymbol{\Psi} \sim \Phi}{\mathbb{E}} \left\| \mathbf{H}'\boldsymbol{\Psi} - \hat{\mathbf{P}}\mathbf{H}\boldsymbol{\Psi} \right\|^2 \leq \underset{\boldsymbol{\Psi} \sim \Phi}{\mathbb{E}} \left\| \mathbf{H}' - \hat{\mathbf{P}}\mathbf{H} \right\|^2 \left\| \boldsymbol{\Psi} \right\|^2. \tag{2}$$

**Remark.** The reformulated distribution matching objective aims to align the essential semantic information between the sample graph and its task-prototype. We further aggregate the representations of the original graph via pooling to compute a trivial closed-form solution $\mathcal{P}^{(t)} = \mathbf{H}' = \texttt{Pooling}(\mathbf{H})$, which serves as the task-prototype reflecting the distribution of the current task.

During the continual learning process, task-prototypes for all tasks are stored in a task-level prototype dictionary $D = \{\mathcal{P}^1, \dots, \mathcal{P}^T\}$. This dictionary is only utilized during inference and does not participate in rehearsal for subsequent training. Specifically, when given a test graph $G^{\text{test}}$ at test time, we predict its task ID by querying the task-prototype dictionary as $\hat{t} = \arg\min(\|\mathcal{P}^{\text{test}} - \mathcal{P}^1\|, \dots, \|\mathcal{P}^{\text{test}} - \mathcal{P}^T\|)$, where $\|\cdot\|$ represents Euclidean distance and $\hat{t}$ is the predicted task ID.

**Class-prototypes for Intra-Task Discrimination.** With each class in the same task sharing the same task-prototype, it is still necessary to enhance the discrimination of classes inside each task. To this end, we introduce class-prototypes as dynamic biases that adaptively refine node representations. Formally, for the feature $\mathbf{X}_i^t \in \mathbf{X}^t$ of the $i$-th node, we compute

$$\Delta_i^t = \mathbf{S}_i^t \Lambda^t, \quad \mathbf{S}_i^t = \texttt{Softmax}(\psi(\mathbf{X}_i^t)), \tag{3}$$

where $\mathbf{S}_i^t$ is the intra-task assignment weight vector and $\psi(\cdot)$ is a linear projection. The class-prototype list $\Lambda^t$ contains $|y^t|$ learnable vectors corresponding to the classes of the current task. By augmenting the GNN encoder representations with gated refinements $\bar{\mathbf{Z}}_i^t = \mathbf{Z}_i^t + \Delta_i^t$, Graph2Hyper dynamically pulls together nodes from the same class and pushes apart nodes from different classes, thereby enhancing class-level discrimination while retaining task-level context.

## 4.2 HYPERNETWORK-BASED PARAMETER-EFFICIENT ADAPTATION

In the GCIL setting, as the number of tasks increases, training a separate classifier head from-scratch for each task incurs significant training overhead. Since node classification across different tasks often exhibits shared characteristics, this motivates us to design a parameter-efficient framework that enables cross-task sharing of model parameters. In particular, instead of updating all parameters, it suffices to generate task-specific modifications conditioned on the characteristics of the current task.

**Adaptive Parameters Generation.** To capture both the commonalities and distinctions among a sequence of tasks, we utilize a hypernetwork to dynamically generate the parameters of task-specific classifiers on the fly, while adapting only part of the hypernetwork for each new task. Concretely, our hypernetwork consists of two linear layers: a frozen, task-shared layer that preserves cross-task knowledge, and a trainable, task-specific layer that captures the unique properties of each task.

We denote the frozen pretrained GNN encoder as a fixed feature extractor $\phi : G^{(t)} \to \mathbb{R}^d$. For the $t$-th task, the continually adapted hypernetwork parameters are given by $\tilde{\Theta}_t = \mathcal{A}_t(\Theta_{t-1})$ or $\tilde{\Theta}_t = \mathrm{Hyper}(\mathbf{Z}^t, \Theta_{t-1})$, where $\mathcal{A}_t(\cdot)$ denotes the adaptation operator on task $t$, $\mathrm{Hyper}(\cdot, \cdot)$ denotes the hypernetwork, and $\mathbf{Z}^t$ is the task embedding. Let the classification head be parameterized by $\Theta \in \mathbb{R}^d$, with the classifier defined as $\hat{h}_\Theta(G^{(t)}) = \langle \Theta, \phi(G^{(t)}) \rangle$. The classifier inherited from the $(t-1)$-th task is denoted by $\Theta_{t-1}$, with shorthand $\hat{h}_{t-1}(x) = \hat{h}_{\Theta_{t-1}}(x)$. The independently trained classifier on task $t$ is denoted by $\Theta_t^\star$, with $\hat{h}_t^\star(x) = \hat{h}_{\Theta_t^\star}(x)$. By sharing a subset of hypernetwork parameters across tasks and performing lightweight adaptation for new tasks, we explicitly enable cross-task knowledge transfer while preserving shared knowledge.

**Hypernetwork-Based Knowledge Transfer with Tighter Upper Bounds.** We theoretically show that knowledge transfer through hypernetwork parameter sharing is equivalent to transferring knowledge between classifier heads, and further prove that hypernetwork-based adaptation leads to a tighter generalization error upper bound in continual learning.

**Theorem 4.2** (Necessary and Sufficient Equivalence in Parameter Space). Suppose the classification head is linear, i.e., $\hat{h}_\Theta(x) = \langle \Theta, \phi(x) \rangle$. Then the following statements are equivalent:

1. There exist scalars $w_{t-1}, w_t$ (depending on task $t$) such that for all $x$,

$$\hat{h}_{\tilde{\Theta}_t}(x) = w_{t-1} \, \hat{h}_{t-1}(x) + w_t \, \hat{h}_t^\star(x). \tag{4}$$

2. There exist scalars $w_{t-1}, w_t$ (depending on task $t$) such that

$$\tilde{\Theta}_t = \mathcal{A}_t(\Theta_{t-1}) = w_{t-1} \, \Theta_{t-1} + w_t \, \Theta_t^\star. \tag{5}$$

Moreover, if $\{\phi(x)\}$ spans $\mathbb{R}^d$, then statement 1 implies statement 2; conversely, statement 2 always implies statement 1.

**Remark.** Theorem 4.2 provides a necessary and sufficient condition: when the adapted hypernetwork output $\tilde{\Theta}_t$ lies in the affine span $\mathrm{Aff}\{\Theta_{t-1}, \Theta_t^\star\}$, the functional behavior is exactly equivalent to the interpolation in Eq. (4). Conversely, if functional interpolation holds globally, the parameters must be an affine combination. In other words, the adapted $\tilde{\Theta}_t$ and the classification head that has transferred cross-task knowledge are equivalent.

**Theorem 4.3** (First-Order Equivalence and Second-Order Remainder Bound). Suppose that for all $x$, the prediction function $\hat{h}_\Theta(x)$ is twice differentiable with respect to $\Theta$. Let $J_{t-1}(x) = \nabla_\Theta \hat{h}_\Theta(x)\big|_{\Theta_{t-1}} \in \mathbb{R}^d$ denote the Jacobian evaluated at $\Theta_{t-1}$, and let $H_\xi(x)$ be the Hessian at some intermediate point $\xi$ on the segment between $\Theta_{t-1}$ and $\Theta_t^\star$. Assume that the spectral norm of the Hessian is bounded: $\|H_\xi(x)\|_2 \leq M_x$ for all $x$. Define the parameter difference $\Delta_t = \Theta_t^\star - \Theta_{t-1}$, and for any $a \in [0,1]$, we set

$$\tilde{\Theta}_t = \Theta_{t-1} + a \, \Delta_t. \tag{6}$$

Then for all $x$, the prediction at the adapted parameter $\tilde{\Theta}_t$ can be decomposed as

$$\hat{h}_{\tilde{\Theta}_t}(x) = (1-a) \, \hat{h}_{t-1}(x) + a \, \hat{h}_t^\star(x) + \mathcal{R}_t(x), \quad |\mathcal{R}_t(x)| \leq \tfrac{1}{2} M_x \, a(1-a) \, \|\Delta_t\|_2^2, \tag{7}$$

where the remainder term $\mathcal{R}_t(x)$ arises from the second-order expansion. In particular, when the classification head is linear, $M_x = 0$, the remainder vanishes, and the equality reduces to an exact interpolation between $\hat{h}_{t-1}$ and $\hat{h}_t^\star$.

Let the loss be $\ell(y, \hat{h}(x))$, convex in its second argument and $\rho$-Lipschitz. Define the expected risk $\epsilon_t(\hat{h}) = \mathbb{E}_{(x,y) \sim \mathcal{D}_t}[\ell(y, \hat{h}(x))]$. Let

$$\hat{h}_a(x) = (1-a)\hat{h}_{t-1}(x) + a\hat{h}_t^\star(x), \quad \tilde{h}(x) = \hat{h}_{\tilde{\Theta}_t}(x), \tag{8}$$

where $\tilde{\Theta}_t$ is given by Eq. (6). The transfer from interpolation to a risk bound is stated in Theorem 4.4:

**Theorem 4.4** (Risk Transfer Bound from Interpolation to Adaptation). Under the conditions of Theorem 4.3, for any $a \in [0,1]$, we have

$$\epsilon_t(\tilde{h}) \leq (1-a) \, \epsilon_t(\hat{h}_{t-1}) + a \, \epsilon_t(\hat{h}_t^\star) + \rho \, \mathbb{E}_{x \sim \mathcal{D}_t}\big[|\mathcal{R}_t(x)|\big], \tag{9}$$

where $\mathcal{R}_t(x)$ is given in Eq. (7). When the head is linear, $\mathcal{R}_t(x) = 0$, and Eq. (9) reduces to a strict convex combination bound.

**Remark.** By Theorem 4.2 and Theorem 4.3, if the hypernetwork-generated parameters satisfy the form $\tilde{\boldsymbol{\Theta}}_t \doteq (1 - a)\boldsymbol{\Theta}_{t-1} + a\boldsymbol{\Theta}_t^\star$ (equality for linear heads, first-order approximation for general nonlinear heads), then at the functional level the model is equivalent to interpolating between task-specific heads. This shows that **hypernetwork-based adaptation with cross-task knowledge sharing is effectively equivalent to transferring the capability of classification heads across tasks**, with risk behavior controlled by Eq. (9). (Proofs are provided in Appendix C.2, C.3 and C.4.)

**Lemma 4.5** (Error Upper Bound without Cross-Task Knowledge Transfer). Let $\tilde{\mathbb{P}}_{\mathcal{S}}$ and $\tilde{\mathbb{P}}_{\mathcal{T}}$ denote the induced distribution over the feature space for each distribution $\mathbb{P}_{\mathcal{S}}$ and $\mathbb{P}_{\mathcal{T}}$ over the original input space. The following inequality holds for the risk $\epsilon_0^{\mathcal{T}}(\hat{h})$ with single step adaptation on the target distribution $\mathbb{P}_{\mathcal{T}}$:

$$\epsilon_0^{\mathcal{T}}(\hat{h}) \leq \min\{\epsilon^{\mathcal{S}}(h^{\mathcal{S}}, h^{\mathcal{T}}), \epsilon^{\mathcal{T}}(h^{\mathcal{S}}, h^{\mathcal{T}})\} + \epsilon^{\mathcal{S}}(\hat{h}) + d_{\mathcal{H}}(\tilde{\mathbb{P}}_{\mathcal{S}}, \tilde{\mathbb{P}}_{\mathcal{T}}), \tag{10}$$

where $d_{\mathcal{H}}$ represents the $\mathcal{H}$-divergence.

**Theorem 4.6** (Error Upper Bound with Continual Cross-Task Knowledge Transfer). Let $\tilde{\mathbb{P}}_{G_{t-1}}$ and $\tilde{\mathbb{P}}_{G^t}$ denote the induced feature distributions of two consecutive target graphs at steps $t-1$ and $t$. The risk $\epsilon_t(\hat{h})$ with state-evolving prototypes (with mixed weight $w_{t-1} + w_t = 1$, $w_{t-1}, w_t \geq 0$, and $\hat{h} = w_{t-1}\hat{h}_{t-1} + w_t\hat{h}_t$) is upper bounded by:

$$\epsilon_t(\hat{h}) \leq \epsilon_t(\hat{h}, h_{t-1}) + \epsilon_t(h_{t-1}, h_t). \tag{11}$$

**Remark.** According to Lemma 4.5 and Theorem 4.6, we conclude that the error bound $\epsilon_t(\hat{h})$ under continual cross-task knowledge transfer is **tighter** than that obtained without transfer. Detailed proofs and comparisons of the error bounds are provided in Appendix C.5.

**Optimization Objective of Hypernetwork.** For the $t$-th task, the continually adapted hypernetwork parameters are optimized by minimizing a node classification loss:

$$\min_{\boldsymbol{\Theta}_t} \mathbb{E}_{G^t \sim Y^t} \ell_{\text{CE}}(\hat{y}_i^t, y_i^t), \quad \hat{y}_i^t = h_{\boldsymbol{\Theta}_t} \circ [\phi_{\boldsymbol{\Psi}}(\mathbf{X}_i, \mathbf{A}_i) + \Delta_i^t], \tag{12}$$

where $y_i^t$ is the ground-truth label of node $i$ from $G^t$, $\hat{y}_i^t \in \hat{Y}^t$ is the predicted label, and $\ell_{\text{CE}(\cdot)}$ is a cross-entropy loss.

## 5 EXPERIMENTS

In this section, we empirically evaluate the effectiveness of the proposed Graph2Hyper.[1] In particular, we aim to investigate the following research questions:

- **RQ1:** How effective is Graph2Hyper compared with existing baselines under GCIL setting?
- **RQ2:** What are the impacts of different components of Graph2Hyper on the effectiveness?
- **RQ3:** How does Graph2Hyper perform under other task formulations?
- **RQ4:** How sensitive is Graph2Hyper to its hyperparameters?

### 5.1 EXPERIMENTAL SETTINGS

**Datasets.** Following the GCLB (Zhang et al., 2022a), four public graph datasets are employed, including CoraFull (McCallum et al., 2000), Arxiv (Hu et al., 2020), Reddit (Hamilton et al., 2017) and Products (Hu et al., 2020). Specifically, CoraFull and Arxiv are citation networks, Reddit is derived from posts on the Reddit platform, and Products is a co-purchasing network extracted from Amazon. For all datasets, each task is defined to include exactly two classes (Zhang et al., 2022a). Moreover, for each class, the data are split into training, validation, and testing sets with proportions of 0.6, 0.2, and 0.2, respectively.

**Baselines.** We compare our Graph2Hyper against two categories of SOTA continual learning baselin approaches: ❶ General CIL baseline methods: EWC (Kirkpatrick et al., 2017), LwF (Li & Hoiem,

---

[1] The code of Graph2Hyper is available after acceptance.

Table 1: Performance (% ± standard deviation) under the GCIL setting on four large datasets. The results of baselines are derived from the published works. "↑" denotes the higher value represents better performance. Oracle Model can get access to the data of all tasks and task IDs, *i.e.*, it obtains the upper bound performance. "✓" in Replay indicates the use of data rehearsal in the model, and "×" denotes no rehearsal involved. 'NA' is short for Not Applicable. The best results are shown in **bold** and the runner-ups are underlined (the same for tables below).

| Methods | Replay | CoraFull | | Arxiv | | Reddit | | Products | |
|---|---|---|---|---|---|---|---|---|---|
| | | AA/% ↑ | AF/% ↑ | AA/% ↑ | AF/% ↑ | AA/% ↑ | AF/% ↑ | AA/% ↑ | AF/% ↑ |
| Fine-tune | × | 3.5±0.5 | -95.2±0.5 | 4.9±0.0 | -89.7±0.4 | 5.9±1.2 | -97.9±3.3 | 7.6±0.7 | -88.7±0.8 |
| Joint | × | 81.2±0.4 | NA | 51.3±0.5 | NA | 97.1±0.1 | NA | 71.5±0.1 | NA |
| Oracle Model | × | 95.5±0.2 | NA | 90.3±0.4 | NA | 99.5±0.0 | NA | 95.3±0.8 | NA |
| EWC | × | 52.6±8.2 | -38.5±12.1 | 8.5±1.0 | -69.5±8.0 | 10.3±11.6 | -33.2±26.1 | 23.8±3.8 | -21.7±7.5 |
| MAS | × | 6.5±1.5 | -92.3±1.5 | 4.8±0.4 | -72.2±4.1 | 9.2±14.5 | -23.1±28.2 | 16.7±4.8 | -57.0±31.9 |
| GEM | × | 8.4±1.1 | -88.4±1.4 | 4.9±0.0 | -89.8±0.3 | 11.5±5.5 | -92.4±5.9 | 4.5±1.3 | -94.7±0.4 |
| LwF | × | 33.4±1.6 | -59.6±2.2 | 9.9±12.1 | -43.6±11.9 | 86.6±1.1 | -9.2±1.1 | 48.2±1.6 | -18.6±1.6 |
| TWP | × | 62.6±2.2 | -30.6±4.3 | 6.7±1.5 | -50.6±13.2 | 8.0±5.2 | -18.8±9.0 | 14.1±4.0 | -11.4±2.0 |
| ERGNN | ✓ | 34.5±4.4 | -61.6±4.3 | 21.5±5.4 | -70.0±5.5 | 82.7±0.4 | -17.3±0.4 | 48.3±1.2 | -45.7±1.3 |
| SSM-uniform | ✓ | 73.0±0.3 | -14.8±0.5 | 47.1±0.5 | -11.7±1.5 | 94.3±0.1 | -1.4±0.1 | 62.0±1.6 | -9.9±1.3 |
| SSM-degree | ✓ | 75.4±0.1 | -9.7±0.0 | 48.3±0.5 | -10.7±0.3 | 94.4±0.0 | -1.3±0.0 | 63.3±0.1 | -9.6±0.3 |
| SEM-curvature | ✓ | 77.7±0.8 | -10.0±1.2 | 49.9±0.6 | -8.4±1.3 | 96.3±0.1 | -0.6±0.1 | 65.1±1.0 | -9.5±0.8 |
| CaT | ✓ | 80.4±0.5 | -5.3±0.4 | 48.2±0.4 | -12.6±0.7 | 97.3±0.1 | -0.4±0.0 | 70.3±0.9 | -4.5±0.8 |
| DeLoMe | ✓ | 81.0±0.2 | -3.3±0.3 | 50.6±0.3 | 5.1±0.4 | 97.4±0.1 | -0.1±0.1 | 67.5±0.7 | -17.3±0.3 |
| OODCIL | ✓ | 71.3±0.5 | -1.1±0.1 | 19.3±1.4 | -1.0±0.4 | 79.3±0.8 | -0.1±0.0 | 41.6±0.9 | -1.6±0.4 |
| DMSG | ✓ | 77.8±0.3 | -0.5±0.5 | 50.7±0.4 | -1.9±1.0 | 98.1±0.0 | 0.9±0.1 | 66.0±0.4 | -0.9±1.6 |
| TPP | × | 93.4±0.4 | **0.0±0.0** | 85.4±0.1 | **0.0±0.0** | 99.5±0.0 | **0.0±0.0** | 94.0±0.5 | **0.0±0.0** |
| **Graph2Hyper** | × | **94.8±0.6** | **0.0±0.0** | **87.2±0.7** | **0.0±0.0** | **99.5±0.1** | **0.0±0.0** | **94.7±1.7** | **0.0±0.0** |

2017), GEM (Lopez-Paz & Ranzato, 2017) and MAS (Aljundi et al., 2018); ❷ Graph CIL baseline methods: ERGNN (Zhou & Cao, 2021), TWP (Liu et al., 2021), SSM (Zhang et al., 2022c), SEM (Zhang et al., 2023c), CaT (Liu et al., 2023) and DeLoMe (Niu et al., 2024a) and TPP (Niu et al., 2024b). In addition, we include two further baselines: **Fine-Tune** and **Joint**. The Fine-Tune baseline incrementally fine-tunes the model from previous tasks without employing any continual learning strategies. The Joint baseline is an oracle model that has access to all graphs simultaneously and performs GCL across the full dataset of all tasks. We also report the performance of an enhanced **Oracle Model**, which extends the Joint baseline by assuming access to the true task ID of every test sample during inference.

**Evaluation and Implementation.** We report the performance matrix $M \in \mathbb{R}^{T \times T}$, a lower triangular matrix where $M_{i,j}$ (for $i \geq j$) denotes the performance on task $j$ after evaluating on task $i$. To evaluate the performance of continual learning, we adopt two commonly used metrics, Average Accuracy (AA) as $AA = \frac{1}{T} \sum_{i=1}^{T} M_{T,i}$ and Average Forgetting (AF) as $AF = \frac{1}{T-1} \sum_{i=1}^{T-1} (M_{T,i} - M_{i,i})$. The proposed method is implemented under the GCLB library (Zhang et al., 2022a). Following the same setting in TPP (Niu et al., 2024b), Graph2Hyper adopts a two-layer SGC (Wu et al., 2019) as backbone with the same parameters in SEM (Zhang et al., 2023c). The pre-trained GNN encoder and the hypernetwork are both randomly initialized on the first task, and the GNN encoder is frozen after a single round of unsupervised pretraining without further parameter updates. All reported results are averaged over 5 runs with standard deviations.

## 5.2 OVERALL PERFORMANCE COMPARISON (RQ1)

To answer RQ1, we compare Graph2Hyper with representative baselines in Table 1. Following the GCIL setting, Graph2Hyper **achieves the best performance across all four datasets**. Specifically, we can draw the following key observations. ❶ As shown by the Fine-Tune results, directly fine-tuning the model learned from previous tasks on the current task data leads to severe performance degradation, since the knowledge from earlier tasks is easily overwritten by the new tasks. ❷ Among the existing baselines, CIL methods originally proposed for Euclidean data generally fail to achieve satisfactory performance on GCIL, which confirms that the unique properties of graph data must be taken into account. In addition, replay-based methods consistently outperform other baselines, demonstrating the effectiveness of using an external memory buffer to mitigate catastrophic forgetting.

Table 2: Ablation study results of Graph2Hyper and its variants. The best results are **bold** and the runner-ups are underlined.

| Methods | CoraFull | | Arxiv | | Reddit | | Products | |
|---|---|---|---|---|---|---|---|---|
| | AA/% ↑ | AF/% ↑ | AA/% ↑ | AF/% ↑ | AA/% ↑ | AF/% ↑ | AA/% ↑ | AF/% ↑ |
| *w/o Class-P* | 58.55±1.54 | 0.00±0.00 | 64.31±1.84 | 0.00±0.00 | 75.294±3.52 | 0.00±0.00 | 60.16±2.00 | 0.00±0.00 |
| *w/o Task-P* | 2.38±0.06 | -5.32±0.32 | 4.81±0.05 | -4.79±0.95 | 4.98±0.09 | -12.95±0.48 | 4.07±0.07 | -8.35±0.27 |
| *w/o bias* | 56.68±1.39 | 0.00±0.00 | 63.78±1.59 | 0.00±0.00 | 78.81±1.78 | 0.00±0.00 | 58.87±1.04 | 0.00±0.00 |
| *Single head* | 91.31±0.83 | -4.34±1.52 | 76.76±0.99 | -3.84±1.52 | 93.45±2.40 | -8.74±1.38 | 86.87±1.01 | -2.7±1.54 |
| *Freeze 1* | 94.48±0.38 | 0.00±0.00 | 85.19±1.37 | 0.00±0.00 | 99.21±1.79 | 0.00±0.00 | 92.29±1.48 | 0.00±0.00 |
| *Freeze 2* | 94.71±0.25 | 0.00±0.00 | 84.62±1.16 | 0.00±0.00 | 98.92±2.13 | 0.00±0.00 | 90.96±0.42 | 0.00±0.00 |
| *Train Both* | **95.27±0.31** | 0.00±0.00 | 86.71±1.16 | 0.00±0.00 | 99.42±0.62 | 0.00±0.00 | 94.69±1.48 | 0.00±0.00 |
| **Graph2Hyper** | 94.83±0.55 | 0.00±0.00 | **87.18±0.69** | 0.00±0.00 | **99.47±0.06** | 0.00±0.00 | **94.71±1.65** | 0.00±0.00 |

However, these methods still suffer from forgetting, as well as from inter-task separation issues. ❸ Both Graph2Hyper and the baseline TPP are fully forget-free GCIL approaches, achieving an AF value of zero across all four datasets. Both methods also succeed in accurately predicting task IDs. Thanks to the design of task-shared knowledge transfer, Graph2Hyper can outperform TPP in terms of overall average performance, while only requiring the hypernetwork to generate classifier parameters. ❹ The Oracle Model is allowed to access all previously seen data and is provided with task IDs for classification. By relaxing the setting constraints and enlarging the available data, the Oracle Model represents the upper bound of performance under the GCIL setting. Graph2Hyper not only surpasses all baselines but also **achieves comparable AA** to the Oracle Model **with significantly fewer trainable parameters**. ❺ It is worth noting that some replay-based methods yield positive AF values, but this does not imply that our zero forgetting is undesirable. These methods gradually accumulate knowledge of past tasks through rehearsal during subsequent learning, which can even improve performance on earlier tasks.

## 5.3 ABLATION STUDIES (RQ2)

We analyze the effectiveness of the two variants:

- *w/o Class-P*: We remove the Class-Prototype with dynamic bias.
- *w/o Task-P*: We remove the Task-Prototype from task-oriented distribution matching.

Results are presented in Table 2. Removing the class prototype affects intra-task classification: although the model can still predict the correct task ID, the lack of fine-grained discrimination leads to suboptimal performance. In contrast, removing the task prototype directly prevents the model from distinguishing tasks, which causes confusion among all classes under the GCIL setting. Overall, Graph2Hyper outperforms the two variants, *i.e.* "*w/o Class-P*" and "*w/o Task-P*", validating its indispensable effectiveness.

In addition, we further analyze several additional variants:

- *w/o bias*: a finer-grained ablation by removing the dynamic bias term.
- *Single head*: hypernetwork generates a single classifier head for task-specific node classification.
- *Freeze 1*: the first layer of hypernetwork is frozen and does not participate in parameter sharing.
- *Freeze 2*: the second layer of hypernetwork is frozen and does not participate in parameter sharing.
- *Train Both*: both linear layers of hypernetwork are trained from scratch.

Specifically, ❶ without the dynamic bias, the classifier head no longer uses class-prototype information and instead relies directly on the representation of the current sample. Since the class-prototype provides representative information while the raw sample contains additional noise, the performance decreases. However, this does not prevent the classifier head from being generated correctly, so the method still maintains forget-free behavior. ❷ using only a single classifier head for task-specific node classification achieves competitive performance that surpasses many existing baselines, but it cannot fully avoid forgetting. This highlights the effectiveness of Graph2Hyper in addressing adaptation in continual learning. ❸ Freezing either of the hypernetwork's linear layers impairs cross-task knowledge transfer; although the method does not completely fail, its performance is negatively affected. ❹ Training both layers from scratch also achieves strong results. However, given

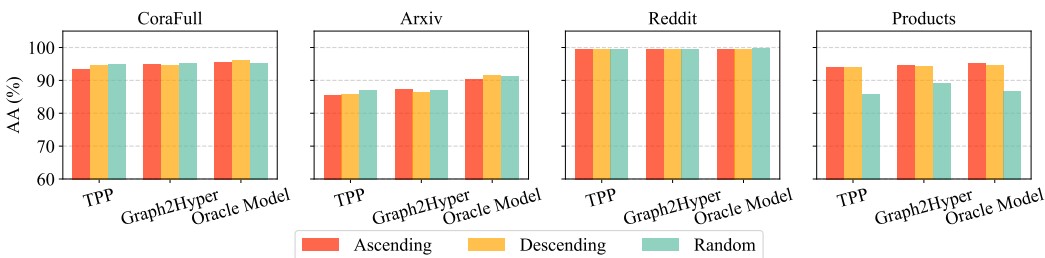

Figure 2: Performance of Graph2Hyper, TPP and Oracle Model under different task orders.

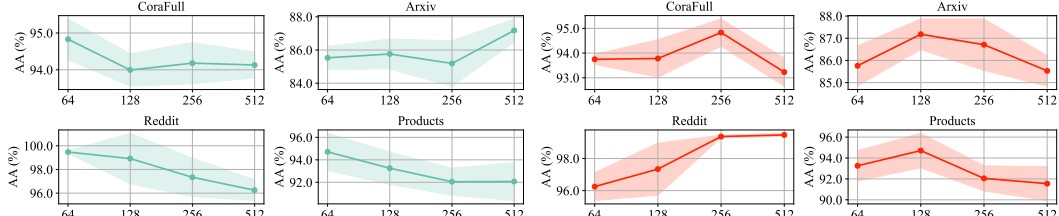

Figure 3: The sensitivity of parameter $dim_1$.  Figure 4: The sensitivity of parameter $dim_2$.

that the current design of Graph2Hyper substantially reduces the number of trainable parameters, it remains more effective and efficient.

## 5.4 ANALYSIS UNDER MORE SETTINGS OF TASK FORMULATION (RQ3)

**Robustness to Different Task Orders.** For task formulation, we assign two distinct node classes to each task and follow the commonly adopted strategy in (Zhang et al., 2022a; Niu et al., 2024a) to ensure fair comparisons with baselines. Specifically, given a graph dataset with multiple classes, we split the classes into tasks in ascending numerical order of their original labels, i.e., classes 0 and 1 form the first task, classes 2 and 3 form the second task, and so forth. To further evaluate the robustness of Graph2Hyper under different task formulations, we additionally construct tasks by splitting classes in two alternative ways: descending numerical order and random ordering of the two classes per task. As illustrated in Figure 2, we observe that under different task orders, our method achieves robustness comparable to the two strongest baselines.

## 5.5 PARAMETER STUDY (RQ4)

**Sensitivity Analysis of Hypernetwork Hidden Dimension.** To analyze the sensitivity of the hypernetwork hidden dimension $dim_1$ in Graph2Hyper, we vary its value within $\{64, 128, 256, 512\}$. The AA with respect to different selections of $dim_1$ is shown in Figure D.1. The results confirm that Graph2Hyper is robust to reasonable choices of $dim_1$, though proper tuning remains important for achieving optimal performance.

**Sensitivity Analysis of GNN Encoder Output Dimension.** We further investigate the effect of the GNN encoder output dimension $dim_2$. As shown in Figure 4, Graph2Hyper again exhibits robustness to reasonable settings of $dim_2$, while tuning still plays a role in obtaining the best performance. Further analysis with more hyperparameters is in Appendix D.9.

## 6 CONCLUSION

In this paper, we present a parameter-efficient framework for graph continual learning, termed **Graph2Hyper**. Concretely, we construct task-level prototypes via distribution matching and further introduce class-level prototypes with dynamic bias to enhance intra-task discrimination. We further propose a lightweight hypernetwork to dynamically generate task-specific classifiers, enabling cross-task knowledge transfer with reduced trainable parameters, with a tighter error bound established for continual optimization. Extensive experiments on multiple benchmarks demonstrate the superiority of Graph2Hyper. Our future work will seek to adapt Graph2Hyper to online settings with streaming data, promoting its scalability and real-world applicability.

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

# Appendix

## Table of Contents

# A  NOTATIONS

As an expansion of the notations in our work, we summarize the frequently used notations in Table A.1.

| Notations | Descriptions |
|---|---|
| $\mathcal{T} = \{G^t\}_{t=1}^{T}$ | Sequence of $T$ tasks (graphs) in GCIL |
| $G^{(t)} = \{\mathbf{A}^{(t)}, \mathbf{X}^{(t)}\}$ | Input graph of the $t$-th task |
| $\mathbf{A}^{(t)} \in \mathbb{R}^{N_t \times N_t}$ | Adjacency matrix of $G^{(t)}$ with $N_t$ nodes |
| $\mathbf{X}^{(t)} \in \mathbb{R}^{N_t \times d}$ | Node feature matrix of $G^{(t)}$ with $d$-dim features |
| $\hat{\mathbf{A}} = \tilde{\mathbf{D}}^{-\frac{1}{2}} \tilde{\mathbf{A}} \tilde{\mathbf{D}}^{-\frac{1}{2}}$ | Normalized adjacency matrix with self-loops |
| $\tilde{\mathbf{A}} = \mathbf{A} + \mathbf{I}$ | Adjacency with self-loops |
| $\tilde{\mathbf{D}}$ | Degree matrix of $\tilde{\mathbf{A}}$ |
| $\mathbf{H} = \hat{\mathbf{A}}^K \mathbf{X}$ | Node embeddings after $K$-hop propagation |
| $\mathbf{H}'$ | Aggregated prototype representation |
| $\hat{\mathbf{P}}$ | Aggregation matrix for task-prototypes |
| $\Psi$ | Learnable linear transformation matrix |
| $\mathbf{Z} = \mathbf{H}\Psi$ | Representation of the original graph |
| $\mathbf{Z}' = \mathbf{H}'\Psi$ | Representation of the task-prototype |
| $\mathcal{P}^{(t)}$ | Task-prototype of the $t$-th task |
| $D = \{\mathcal{P}^1, \ldots, \mathcal{P}^T\}$ | Dictionary of task-prototypes |
| $\Lambda^t$ | Class-prototype list for the $t$-th task |
| $\mathbf{S}_i^t$ | Intra-task assignment weight vector of node $i$ |
| $\Delta_i^t = \mathbf{S}_i^t \Lambda^t$ | Dynamic refinement bias for node $i$ |
| $\bar{\mathbf{Z}}_i^t = \mathbf{Z}_i^t + \Delta_i^t$ | Refined representation of node $i$ |
| $\phi : G^{(t)} \to \mathbb{R}^d$ | Frozen pretrained GNN encoder |
| $\Theta_t$ | Classifier parameters adapted for task $t$ |
| $\Theta_{t-1}$ | Classifier parameters inherited from task $t-1$ |
| $\Theta_t^\star$ | Independently trained classifier for task $t$ |
| $\tilde{\Theta}_t$ | Adapted hypernetwork parameters for task $t$ |
| $\mathbf{Z}^t$ | Task embedding of task $t$ |
| $\hat{h}_\Theta(G^{(t)})$ | Classifier with parameters $\Theta$ |
| $\hat{h}_{t-1}, \hat{h}_t^\star$ | Classifiers from previous and independent training |
| $\epsilon^t(\hat{h})$ | Prediction risk on task $t$ |
| $d_\mathcal{H}(\cdot, \cdot)$ | $\mathcal{H}$-divergence between two distributions |
| $\ell_{\mathrm{CE}}(\cdot)$ | Cross-entropy loss function |
| $y_i^t, \hat{y}_i^t$ | Ground-truth and predicted label of node $i$ in task $t$ |
| $\alpha, \beta, \tau$ | Hyperparameters of Graph2Hyper |

Table A.1: Summary of notations and descriptions.

# B  ALGORITHM

The overall optimization process of our Graph2Hyper is shown in Algorithm 1.

## B.1  COMPLEXITY ANALYSIS

We analyze the time complexity of Graph2Hyper by dividing it into three main components:

- **Task-Prototype Construction**: $\mathcal{O}((|A^t|F + NdF + Nd)E_1)$, where $|A^t|$ is the number of edges, $N$ is the number of nodes, $F$ is the input feature dimension, $d$ is the hidden dimension, and $E_1$ is the number of propagation epochs.

- **Class-Prototype Refinement**: $\mathcal{O}((Nd^2 + Nd)E_2)$, where $E_2$ is the number of training epochs for refining class-prototypes.

- **Hypernetwork-Based Adaptation**: $\mathcal{O}((dC + d^2)E_3)$, where $C$ is the number of classes and $E_3$ is the number of adaptation epochs for the hypernetwork.

---

**Algorithm 1:** Continual Adaptation with Task- and Class-Prototypes (Graph2Hyper).

**Input:** Sequence of task graphs $G^t = \{\mathbf{A}^{(t)}, \mathbf{X}^{(t)}\}_{t=1}^T$;
Pre-trained GNN encoder $\phi$ with frozen parameters;
Hypernetwork with parameters $\boldsymbol{\Theta}_0$
**Output:** Adapted classifiers $\{\boldsymbol{\Theta}_t\}_{t=1}^T$

1   Initialize model with $\boldsymbol{\Theta}_0$;
2   Initialize task-prototype dictionary $D \leftarrow \emptyset$;
3   **for** $t = 1$ *to* $T$ **do**
4      // *Task-Prototype Construction*
5      Propagate features: $\mathbf{H}^{(t)} \leftarrow \hat{\mathbf{A}}^K \mathbf{X}^{(t)}$;
6      Aggregate task-level features: $\mathbf{H}'^{(t)} \leftarrow \texttt{Pooling}(\mathbf{H}^{(t)})$;
7      Define task-prototype $\mathcal{P}^{(t)} \leftarrow \mathbf{H}'^{(t)}$;
8      Update dictionary $D \leftarrow D \cup \{\mathcal{P}^{(t)}\}$;
9      // *Class-Prototype Refinement*
10     **for** *each node $i$ in $G^{(t)}$* **do**
11        Compute assignment weights: $\mathbf{S}_i^t \leftarrow \texttt{Softmax}(\psi(\mathbf{X}_i^t))$;
12        Compute dynamic bias: $\Delta_i^t \leftarrow \mathbf{S}_i^t \Lambda^t$;
13        Refine representation: $\bar{\mathbf{Z}}_i^t \leftarrow \mathbf{Z}_i^t + \Delta_i^t$;
14     **end**
15     // *Hypernetwork-Based Adaptation*
16     Generate task-specific classifier: $\tilde{\boldsymbol{\Theta}}_t \leftarrow \text{Hyper}(\mathbf{Z}^t, \boldsymbol{\Theta}_{t-1})$;
17     Compute cross-entropy loss: $\mathcal{L}_t \leftarrow \mathbb{E}_{G^t} \ell_{\text{CE}}(\hat{y}_i^t, y_i^t)$;
18     Update hypernetwork parameters via gradient descent: $\boldsymbol{\Theta}_{t-1} \rightarrow \boldsymbol{\Theta}_t$;
19   **end**
20   **return** $\{\boldsymbol{\Theta}_t\}_{t=1}^T$

---

Therefore, the overall time complexity across $T$ tasks in GCIL is $\mathcal{O}\Big((|A^1|F + NdF + Nd)E_1 + \sum_{t=1}^T \big[(|A^t|F + NdF + Nd)E_1 + (Nd^2 + Nd)E_2 + (dC + d^2)E_3\big]\Big)$.

## B.2   Parameter Efficiency Analysis

In this subsection, we provide a detailed comparison between the conventional "one-classifier-per-task" strategy and our hypernetwork-based approach. We explicitly quantify the number of trainable parameters required during continual adaptation and show that our method is significantly more parameter-efficient.

**TPP Baseline: One Classifier per Task.** In the baseline setting, each task $t$ is equipped with an independent linear classifier head: $h^{(t)}(x) = \mathbf{W}^{(t)} x + \mathbf{b}^{(t)}$, where $\mathbf{W}^{(t)} \in \mathbb{R}^{C \times d}$ and $\mathbf{b}^{(t)} \in \mathbb{R}^C$. The number of parameters per task is therefore $P_{\text{head}} = dC + C = (d+1)C$. Since the classifiers are not shared across tasks, the overall number of parameters that need to be trained and stored grows linearly with the number of tasks $T$: $P_{\text{baseline,total}} = T(dC + C)$.

**Graph2Hyper: Hypernetwork-based Classifier Generation.** Our hypernetwork consists of two linear layers: $\text{Linear}(e \rightarrow h) \rightarrow \text{ReLU} \rightarrow \text{Linear}(h \rightarrow (dC + C))$, where $e$ is the task embedding dimension, $h$ is the hidden dimension of the hypernetwork, and $(dC + C)$ is the dimension required to generate both $\mathbf{W}$ and $\mathbf{b}$. Importantly, the second layer is *frozen*, so only the first layer is updated across tasks. Thus, the number of trainable parameters per task is $P_{\text{hyper,train}} = eh + h = (e+1)h$. The frozen second layer contributes a constant storage cost of $P_{\text{frozen}} = h(dC + C) + (dC + C)$, which does not increase with the number of tasks $T$.

If task embeddings are learnable, each task adds an additional $e$ parameters, yielding a per-task cost of $(e+1)h + e$. However, this remains significantly smaller than $(d+1)C$ when $e, h \ll d, C$.

**Concrete Comparison.** To make the savings more explicit, consider a typical GCIL setup with hidden dimension $d = 256$, number of classes per task $C = 10$, task embedding dimension $e = 32$, and hypernetwork hidden dimension $h = 64$. Baseline (per-task classifier head): $P_{\text{head}} = (256 + 1) \times 10 = 2{,}570.$, while Graph2Hyper (hypernetwork per-task trainable parameters): $P_{\text{hyper,train}} = (32+1) \times 64 = 2{,}112$. Thus, even with moderate hypernetwork size, our per-task trainable parameters are already smaller than a single classifier head. Moreover, across $T = 50$ tasks, the baseline requires training and storing $50 \times 2{,}570 = 128{,}500$ task-specific parameters, whereas our method only updates the same 2,112 parameters across all tasks (plus optional lightweight embeddings of size 32 per task). The frozen second layer remains constant and independent of $T$.

Compared with the baseline that trains and stores a separate classifier head for each task, our hypernetwork-based approach is substantially more parameter-efficient. The total number of trainable parameters in our framework does not grow linearly with the number of tasks, but remains nearly constant (or increases very slowly with lightweight embeddings). This design choice enables continual learning with far fewer trainable parameters, while still adapting effectively to new tasks.

## C  THEORETICAL JUSTIFICATION

### C.1  PROOF OF TASK-ORIENTED DISTRIBUTION MATCHING

We denote the prototype corresponding to $G^{(t)}$ as $\tilde{G}^{(t)} = \{\mathbf{A}', \mathbf{X}'\}$. To ensure that the extracted prototype faithfully represents the original graph, we reformulate a prototype-oriented distribution matching objective between the representations of the original graph and its prototype.

**Proposition C.1.** After multi-hop propagation, let $\mathbf{Z} = \mathbf{H}\mathbf{\Psi} = \hat{\mathbf{A}}^K \mathbf{X}\mathbf{\Psi}$ and $\mathbf{Z}' = \mathbf{H}'\mathbf{\Psi} = \hat{\mathbf{A}}'^K \mathbf{X}'\mathbf{\Psi}$ be the representations of the original graph $G^{(t)}$ and its prototype $\tilde{G}^{(t)}$, respectively. To directly align the distribution of the input domain with that of its prototype, the discrepancy between the propagated features should be minimized as:

$$\min_{\tilde{G}^{(t)}} \mathbb{E}_{\mathbf{\Psi} \sim \Phi} \left\| \mathbf{P}'\mathbf{H}'\mathbf{\Psi} - \mathbf{P}\mathbf{H}\mathbf{\Psi} \right\|^2. \tag{C.1}$$

**Remark.** The objective is upper-bounded by:

$$\mathbb{E}_{\mathbf{\Psi} \sim \Phi} \left\| \mathbf{H}'\mathbf{\Psi} - \hat{\mathbf{P}}\mathbf{H}\mathbf{\Psi} \right\|^2 \leq \mathbb{E}_{\mathbf{\Psi} \sim \Phi} \left\| \mathbf{H}' - \hat{\mathbf{P}}\mathbf{H} \right\|^2 \|\mathbf{\Psi}\|^2. \tag{C.2}$$

where $\mathbf{\Psi}$ is the learnable weight matrix to transform the $K$-th order propagated features $\mathbf{H}'$ and $\mathbf{H}$. $\hat{\mathbf{A}}' = \tilde{\mathbf{D}}'^{\frac{1}{2}} \tilde{\mathbf{A}}' \tilde{\mathbf{D}}'^{\frac{1}{2}}$ and $\hat{\mathbf{A}} = \tilde{\mathbf{D}}^{\frac{1}{2}} \tilde{\mathbf{A}} \tilde{\mathbf{D}}^{\frac{1}{2}}$ represent the symmetric normalized adjacency matrices, where $\tilde{\mathbf{A}}'$ and $\tilde{\mathbf{A}}$ are adjacency matrices with self-loops. $\tilde{\mathbf{D}}'$ and $\tilde{\mathbf{D}}$ are degree matrices for $\tilde{\mathbf{A}}'$ and $\tilde{\mathbf{A}}$, respectively. Since $\mathbf{\Psi}$ is independent of both $\tilde{G}^{(t)}$ and $\hat{\mathbf{P}}$, we can minimize the upper bound by:

$$\arg \min_{\tilde{G}^{(t)}, \hat{\mathbf{P}}} \left\| \mathbf{H}' - \hat{\mathbf{P}}\mathbf{H} \right\|^2. \tag{C.3}$$

According to Proposition C.1, we expect a symmetric encoding procedure between the original $G^{(t)}$ and domain-specific prototype $\tilde{G}^{(t)}$. Therefore, our objective is to construct $\mathbf{A}'$ and $\mathbf{X}'$ satisfying $\hat{\mathbf{A}}'^K \mathbf{X}' = \mathbf{H}'$. To this end, we utilize the pre-defined graph structure and calculate $\mathbf{X}'$ in a close-formed solution. Specifically, we construct the structure via a similarity-based thresholding scheme:

$$\mathbf{A}'_{i,j} = \begin{cases} 1 & \text{if } \cos(\mathbf{H}'_i, \mathbf{H}'_j) > \tau, \\ 0 & \text{otherwise,} \end{cases} \tag{C.4}$$

where $\cos(\cdot, \cdot)$ measures the cosine similarity and $\tau$ is the hyper-parameter for graph sparsification. To ensure that generated features change smoothly between connected nodes, we introduce the Dirichlet energy constraint Kalofolias (2016) in feature reconstruction loss and quantify the smoothness of graph signals. The optimization objective for $\mathbf{X}'$ is then formulated as:

$$\arg \min_{\mathbf{X}'} \left\| \hat{\mathbf{A}}'^K \mathbf{X}' - \mathbf{H}' \right\|^2 + \alpha \operatorname{tr}(\mathbf{X}'^\top \mathbf{L}' \mathbf{X}'), \tag{C.5}$$

where $\mathbf{L}' = \mathbf{D}' - \mathbf{A}'$ is the Laplacian matrix, $\mathbf{D}'$ is degree matrix, and $\alpha$ controls the smoothness strength.

**Proposition C.2.** Assume that $\tilde{G}^{(t)} = \{\mathbf{A}', \mathbf{X}'\}$ is a prototype of input sample, the closed-form solution of Eq. (C.5) takes the form as $\mathbf{X}' = (\mathbf{Q}^\top \mathbf{Q} + \alpha \mathbf{L}')^{-1} \mathbf{Q}^\top \mathbf{H}'$, where $\mathbf{Q} = \hat{\mathbf{A}}'^K$.

*Proof.* Let us denote the objective in Eq. (C.5) as $\mathcal{I} = \left\| \hat{\mathbf{A}}'^K \mathbf{X}' - \mathbf{H}' \right\|^2 + \alpha \operatorname{tr}(\mathbf{X}'^\top \mathbf{L}' \mathbf{X}')$. To solve the optimization problem in Eq. (C.5), we compute the first- and second-order derivatives of $\mathcal{I}$ with respect to $\mathbf{X}'$:

$$\frac{\partial \mathcal{I}}{\partial \mathbf{X}'} = 2\mathbf{Q}^\top(\mathbf{Q}\mathbf{X}' - \mathbf{H}') + \alpha(\mathbf{L}' + \mathbf{L}'^\top)\mathbf{X}'$$
$$= 2(\mathbf{Q}^\top \mathbf{Q} + \alpha \mathbf{L}')\mathbf{X}' - 2\mathbf{Q}^\top \mathbf{H}', \tag{C.6}$$
$$\frac{\partial^2 \mathcal{I}}{\partial \mathbf{X}'^2} = 2(\mathbf{Q}^\top \mathbf{Q} + \alpha \mathbf{L}').$$

According to the definition of $\mathbf{A}'$ in Eq. (C.4), both $\mathbf{L}'$ and $\mathbf{Q}$ are positive semi-definite matrices. Therefore, the second-order derivative $\frac{\partial^2 \mathcal{I}}{\partial \mathbf{X}'^2}$ is positive semi-definite, and the objective function $\mathcal{I}$ is convex. This implies that the first-order derivative $\frac{\partial \mathcal{I}}{\partial \mathbf{X}'}$ corresponds to a convex function with a unique minimum. Setting the gradient to zero yields the closed-form solution:

$$\mathbf{X}' = (\mathbf{Q}^\top \mathbf{Q} + \alpha \mathbf{L}')^{-1} \mathbf{Q}^\top \mathbf{H}', \tag{C.7}$$

where $\mathbf{Q} = \hat{\mathbf{A}}'^K$. $\square$

Specifically, when the task-prototype is represented as a single vector, $\tilde{G}^{(t)}$ contains only one node, in which case $\mathbf{X}' = \mathbf{H}'$ serves as the closed-form solution of the feature.

## C.2 PROOF OF THEOREM C.3

We denote the frozen pretrained GNN encoder as a fixed feature extractor $\phi : G^{(t)} \to \mathbb{R}^d$. For the $t$-th task, the continually adapted hypernetwork parameters are given by $\tilde{\Theta}_t = \mathcal{A}_t(\Theta_{t-1})$ or $\tilde{\Theta}_t = \operatorname{Hyper}(\mathbf{Z}^t, \Theta_{t-1})$, where $\mathcal{A}_t(\cdot)$ denotes the adaptation operator on task $t$, $\operatorname{Hyper}(\cdot, \cdot)$ denotes the hypernetwork, and $\mathbf{Z}^t$ is the task embedding. A task-specific classification head is parameterized by $\Theta \in \mathbb{R}^d$, with prediction function (classification head) defined as $\hat{h}_\Theta(G^{(t)}) = \langle \Theta, \phi(G^{(t)}) \rangle$. The head from the $(t-1)$-th task is denoted by $\Theta_{t-1}$, with shorthand $\hat{h}_{t-1}(x) = \hat{h}_{\Theta_{t-1}}(x)$. The independently trained head on task $t$ is denoted by $\Theta_t^\star$, with $\hat{h}_t^\star(x) = \hat{h}_{\Theta_t^\star}(x)$. By Theorem C.3, we establish a necessary and sufficient equivalence between hypernetwork parameters with cross-task knowledge transfer and classification-head parameter space.

**Theorem C.3** (Necessary and Sufficient Equivalence in Parameter Space). Suppose the classification head is linear, i.e., $\hat{h}_\Theta(x) = \langle \Theta, \phi(x) \rangle$. Then the following statements are equivalent:

1. There exist scalars $w_{t-1}, w_t$ (depending on task $t$) such that for all $x$,
$$\hat{h}_{\tilde{\Theta}_t}(x) = w_{t-1} \hat{h}_{t-1}(x) + w_t \hat{h}_t^\star(x). \tag{C.8}$$

2. There exist scalars $w_{t-1}, w_t$ (depending on task $t$) such that
$$\tilde{\Theta}_t = \mathcal{A}_t(\Theta_{t-1}) = w_{t-1} \Theta_{t-1} + w_t \Theta_t^\star. \tag{C.9}$$

Moreover, if $\{\phi(x)\}$ spans $\mathbb{R}^d$, then statement 1 implies statement 2; conversely, statement 2 always implies statement 1.

*Proof.* (2⇒1) By linearity we have
$$\hat{h}_{\tilde{\Theta}_t}(x) = \langle w_{t-1}\Theta_{t-1} + w_t\Theta_t^\star, \phi(x) \rangle = w_{t-1}\hat{h}_{t-1}(x) + w_t\hat{h}_t^\star(x). \tag{C.10}$$

(1⇒2) Assume that for all $x$,
$$\langle \tilde{\Theta}_t, \phi(x) \rangle = \langle w_{t-1}\Theta_{t-1} + w_t\Theta_t^\star, \phi(x) \rangle. \tag{C.11}$$

Let $\mathbf{v} = \tilde{\Theta}_t - (w_{t-1}\Theta_{t-1} + w_t\Theta_t^\star)$. Then $\langle \mathbf{v}, \phi(x) \rangle = 0$ for all $x$. If $\{\phi(x)\}$ spans $\mathbb{R}^d$, this implies that $\mathbf{v}$ is orthogonal to every vector in $\mathbb{R}^d$, hence $\mathbf{v} = \mathbf{0}$. Therefore, Eq. (C.9) holds. $\square$

**Remark.** Theorem C.3 provides a necessary and sufficient condition: when the adapted hypernetwork output $\tilde{\Theta}_t$ lies in the affine span $\text{Aff}\{\Theta_{t-1}, \Theta_t^\star\}$, the functional behavior is exactly equivalent to the interpolation in Eq. (C.8). Conversely, if functional interpolation holds globally, the parameters must be an affine combination. In other words, the adapted $\tilde{\Theta}_t$ and the classification head that has transferred cross-task knowledge are equivalent.

By Lemma C.4, we provide a closed-form example of an adaptation operator with interpretable weights as follows:

**Lemma C.4** (Affine Form of One Gradient Step and One Proximal Step). Let the squared loss for task $t$ be $\mathcal{L}_t(\Theta) = \frac{1}{2}\|\Phi_t\Theta - y_t\|_2^2$, where $\Phi_t = [\phi(x_i)^\top]_i$. Then:

1. (One gradient step) Starting from $\Theta_{t-1}$ with learning rate $\eta > 0$:
$$\tilde{\Theta}_t = \Theta_{t-1} - \eta\,\nabla\mathcal{L}_t(\Theta_{t-1}) = \big(I - \eta\,\Phi_t^\top\Phi_t\big)\Theta_{t-1} + \eta\,\Phi_t^\top y_t. \tag{C.12}$$

2. (One proximal step) Minimizing $\min_\Theta\ \mathcal{L}_t(\Theta) + \frac{\lambda}{2}\|\Theta - \Theta_{t-1}\|_2^2$ yields the closed-form solution
$$\tilde{\Theta}_t = \big(\Phi_t^\top\Phi_t + \lambda I\big)^{-1}\big(\Phi_t^\top y_t + \lambda\Theta_{t-1}\big). \tag{C.13}$$

If the whitening condition $\Phi_t^\top\Phi_t = \alpha_t I$ holds, and the least-squares solution is
$$\Theta_t^\star = (\Phi_t^\top\Phi_t)^{-1}\Phi_t^\top y_t = \alpha_t^{-1}\Phi_t^\top y_t, \tag{C.14}$$

then Eq. (C.12) and Eq. (C.13) respectively reduce to
$$\tilde{\Theta}_t = (1 - \eta\alpha_t)\,\Theta_{t-1} + (\eta\alpha_t)\,\Theta_t^\star, \quad \tilde{\Theta}_t = \tfrac{\lambda}{\alpha_t+\lambda}\,\Theta_{t-1} + \tfrac{\alpha_t}{\alpha_t+\lambda}\,\Theta_t^\star. \tag{C.15}$$

*Proof.* For Eq. (C.12): since $\nabla\mathcal{L}_t(\Theta) = \Phi_t^\top(\Phi_t\Theta - y_t)$, one gradient descent step at $\Theta_{t-1}$ yields the result.

For Eq. (C.13): the optimization objective is a strictly convex quadratic function. Setting the gradient to zero gives
$$\Phi_t^\top(\Phi_t\tilde{\Theta}_t - y_t) + \lambda(\tilde{\Theta}_t - \Theta_{t-1}) = \mathbf{0}. \tag{C.16}$$
Rearranging yields $(\Phi_t^\top\Phi_t + \lambda I)\tilde{\Theta}_t = \Phi_t^\top y_t + \lambda\Theta_{t-1}$, from which Eq. (C.13) follows. Under the whitening condition, substituting $\Theta_t^\star = \alpha_t^{-1}\Phi_t^\top y_t$ into Eq. (C.12) and Eq. (C.13) yields Eq. (C.15). $\square$

By Theorem C.3 together with Eq. (C.15), we conclude that under the linear-head and whitening assumptions, both one gradient step and one proximal adaptation are exactly equivalent to functional interpolation, with weights determined by $(\eta, \alpha_t)$ or $(\lambda, \alpha_t)$.

**Analysis of our hypernetwork architecture.** The hypernetwork is designed to achieve parameter-efficiency and controlled knowledge transfer across tasks. Only the first linear layer is trainable, while the second linear layer is fixed. This structure is chosen because it satisfies the mathematical properties we analyze: the trainable layer captures shared transferable components, and the fixed layer restricts the classifier to a stable parameter space, which is essential for rehearsal-free continual learning.

## C.3 PROOF OF THEOREM C.5

In general, if we relax the restriction on the form of the classification head and allow it to be nonlinear, we only assume that $\hat{h}_\theta(x)$ is differentiable with respect to $\theta$ and has bounded second derivatives.

**Theorem C.5** (First-Order Equivalence and Second-Order Remainder Bound). Suppose that for all $x$, the prediction function $\hat{h}_\Theta(x)$ is twice differentiable with respect to $\Theta$. Let $J_{t-1}(x) = \nabla_\Theta\hat{h}_\Theta(x)\big|_{\Theta_{t-1}} \in \mathbb{R}^d$ denote the Jacobian evaluated at $\Theta_{t-1}$, and let $H_\xi(x)$ be the Hessian at some intermediate point $\xi$ on the segment between $\Theta_{t-1}$ and $\Theta_t^\star$. Assume that the spectral norm of the Hessian is bounded: $\|H_\xi(x)\|_2 \le M_x$ for all $x$. Define the parameter difference $\Delta_t = \Theta_t^\star - \Theta_{t-1}$, and for any $a \in [0, 1]$, we set
$$\tilde{\Theta}_t = \Theta_{t-1} + a\,\Delta_t. \tag{C.17}$$

Then for all $x$, the prediction at the adapted parameter $\tilde{\boldsymbol{\Theta}}_t$ can be decomposed as

$$\hat{h}_{\tilde{\boldsymbol{\Theta}}_t}(x) = (1-a)\,\hat{h}_{t-1}(x) + a\,\hat{h}_t^\star(x) + \mathcal{R}_t(x), \quad |\mathcal{R}_t(x)| \leq \tfrac{1}{2}\,M_x\,a(1-a)\,\|\Delta_t\|_2^2, \quad \text{(C.18)}$$

where the remainder term $\mathcal{R}_t(x)$ arises from the second-order expansion. In particular, when the classification head is linear, $M_x = 0$, the remainder vanishes, and the equality reduces to an exact interpolation between $\hat{h}_{t-1}$ and $\hat{h}_t^\star$.

*Proof.* Apply a second-order Taylor expansion around $\boldsymbol{\Theta}_{t-1}$:

$$\hat{h}_{\boldsymbol{\Theta}_{t-1}+a\Delta_t}(x) = \hat{h}_{t-1}(x) + a\,\langle J_{t-1}(x), \Delta_t\rangle + \tfrac{1}{2}a^2\,\Delta_t^\top H_{\xi_a}(x)\Delta_t, \quad \text{(C.19)}$$

where $J_{t-1}(x) = \nabla_{\boldsymbol{\Theta}}\hat{h}_{\boldsymbol{\Theta}}(x)\big|_{\boldsymbol{\Theta}_{t-1}}$, and $H_{\xi_a}(x)$ is the Hessian at some point along the segment. Similarly,

$$\hat{h}_t^\star(x) = \hat{h}_{\boldsymbol{\Theta}_{t-1}+\Delta_t}(x) = \hat{h}_{t-1}(x) + \langle J_{t-1}(x), \Delta_t\rangle + \tfrac{1}{2}\,\Delta_t^\top H_{\xi_1}(x)\Delta_t. \quad \text{(C.20)}$$

Eliminating the first-order terms yields

$$\hat{h}_{\boldsymbol{\Theta}_{t-1}+a\Delta_t}(x) = (1-a)\hat{h}_{t-1}(x) + a\hat{h}_t^\star(x) + \tfrac{1}{2}\Big(a^2 H_{\xi_a}(x) - a H_{\xi_1}(x)\Big):(\Delta_t\Delta_t^\top). \quad \text{(C.21)}$$

Taking the spectral norm bound of the last term, and noting that $a^2 - a = -a(1-a)$, gives

$$|\mathcal{R}_t(x)| \leq \tfrac{1}{2}\,a(1-a)\,\|\Delta_t\|_2^2\,\max\{\|H_{\xi_a}(x)\|_2, \|H_{\xi_1}(x)\|_2\} \leq \tfrac{1}{2}\,M_x\,a(1-a)\,\|\Delta_t\|_2^2. \quad \text{(C.22)}$$

$\square$

**Further Analysis.** Theorem C.5 proves that changes in the generated classifier are bounded by changes in the task embedding. The fixed second linear layer provides this stability by acting as a contraction map. This is why we freeze the second layer which ensures that classifier parameters remain stable across tasks and prevents uncontrolled drift.

Let the loss be $\ell(y, \hat{h}(x))$, convex in its second argument and $\rho$-Lipschitz. Define the expected risk $\epsilon_t(\hat{h}) = \mathbb{E}_{(x,y)\sim\mathcal{D}_t}[\ell(y, \hat{h}(x))]$.

Let

$$\hat{h}_a(x) = (1-a)\hat{h}_{t-1}(x) + a\hat{h}_t^\star(x), \quad \tilde{h}(x) = \hat{h}_{\tilde{\boldsymbol{\Theta}}_t}(x), \quad \text{(C.23)}$$

where $\tilde{\boldsymbol{\Theta}}_t$ is given by Eq. (C.17). The transfer from interpolation to a risk bound is stated in Theorem C.6.

### C.4 PROOF OF THEOREM C.6

**Theorem C.6** (Risk Transfer Bound from Interpolation to Adaptation). Under the conditions of Theorem C.5, for any $a \in [0, 1]$, we have

$$\epsilon_t(\tilde{h}) \leq (1-a)\,\epsilon_t(\hat{h}_{t-1}) + a\,\epsilon_t(\hat{h}_t^\star) + \rho\,\mathbb{E}_{x\sim\mathcal{D}_t}\big[|\mathcal{R}_t(x)|\big], \quad \text{(C.24)}$$

where $\mathcal{R}_t(x)$ is given in Eq. (C.18). When the head is linear, $\mathcal{R}_t(x) = 0$, and Eq. (C.24) reduces to a strict convex combination bound.

*Proof.* By convexity,

$$\ell\big(y, \hat{h}_a(x)\big) \leq (1-a)\ell\big(y, \hat{h}_{t-1}(x)\big) + a\,\ell\big(y, \hat{h}_t^\star(x)\big). \quad \text{(C.25)}$$

By Lipschitz continuity,

$$\ell\big(y, \tilde{h}(x)\big) \leq \ell\big(y, \hat{h}_a(x)\big) + \rho\,|\tilde{h}(x) - \hat{h}_a(x)|. \quad \text{(C.26)}$$

By Theorem C.5, $\tilde{h}(x) - \hat{h}_a(x) = \mathcal{R}_t(x)$. Taking expectations yields Eq. (C.24). $\square$

**Remark.** By Theorem C.3 and Theorem C.5, if the hypernetwork-generated parameters satisfy the form $\tilde{\Theta}_t \doteq (1 - a)\,\Theta_{t-1} + a\,\Theta_t^\star$ (equality for linear heads, first-order approximation for general nonlinear heads), then at the functional level the model is equivalent to interpolating between task-specific heads. This shows that **hypernetwork-based adaptation with cross-task knowledge sharing is effectively equivalent to transferring the capability of classification heads across tasks**, with risk behavior controlled by Eq. (C.24).

**Further Analysis.** Theorem C.6 establishes that the affine combination induced by the hypernetwork yields a structured transfer term between tasks, determined by the shared space and embedding geometry. This motivates our choice of a low-dimensional task embedding and a single shared trainable layer, which together control the extent of cross-task influence and avoid interference.

## C.5 PROOF OF ERROR UPPER BOUNDS

**Definition C.7** (Classification error)**.** The classification error of the function $\hat{h}$ under task $\mathcal{D}_i$ is defined as

$$\epsilon_i(\hat{h}) = \mathbb{E}_{X \sim \mathcal{D}_i}|\hat{h}(X) - h(X)|. \tag{C.27}$$

For binary classification functions $h$ and $\hat{h}$, we have:

$$
\begin{aligned}
\epsilon_i(\hat{h}) = \epsilon_i(\hat{h}, h) &= \mathbb{E}_{X \sim \mathcal{D}_i}|\hat{h}(X) - h(X)| \\
&= \mathbb{P}_{X \sim \mathcal{D}_i}(\hat{h}(X) \neq h(X)).
\end{aligned}
\tag{C.28}
$$

**Definition C.8** ($\mathcal{H}$-divergence)**.** Given two induced task feature space distributions $\tilde{\mathcal{D}}_s, \tilde{\mathcal{D}}_t$ and a hypothesis space $\mathcal{H}$, the $\mathcal{H}$-divergence between $\tilde{\mathcal{D}}_s$ and $\tilde{\mathcal{D}}_t$ is defined as:

$$d_{\mathcal{H}}(\tilde{\mathcal{D}}_s, \tilde{\mathcal{D}}_t) = 2 \sup_{h \in \mathcal{H}} \left|\mathbb{E}_{X \sim \tilde{\mathcal{D}}_s}[h(X) = 1] - \mathbb{E}_{X \sim \tilde{\mathcal{D}}_t}[h(X) = 1]\right|. \tag{C.29}$$

**Lemma C.9** (Error Upper Bound without Cross-Task Knowledge Transfer (Zhao et al., 2019))**.** Let $\tilde{\mathbb{P}}_{\mathcal{S}}$ and $\tilde{\mathbb{P}}_{\mathcal{T}}$ denote the induced distribution over the feature space for each distribution $\mathbb{P}_{\mathcal{S}}$ and $\mathbb{P}_{\mathcal{T}}$ over the original input space. The following inequality holds for the risk $\epsilon_0^{\mathcal{T}}(\hat{h})$ with single step adaptation on the target distribution $\mathbb{P}_{\mathcal{T}}$:

$$\epsilon_0^{\mathcal{T}}(\hat{h}) \leq \min\{\epsilon^{\mathcal{S}}(h^{\mathcal{S}}, h^{\mathcal{T}}), \epsilon^{\mathcal{T}}(h^{\mathcal{S}}, h^{\mathcal{T}})\} + \epsilon^{\mathcal{S}}(\hat{h}) + d_{\mathcal{H}}(\tilde{\mathbb{P}}_{\mathcal{S}}, \tilde{\mathbb{P}}_{\mathcal{T}}), \tag{C.30}$$

where $d_{\mathcal{H}}$ represents the $\mathcal{H}$-divergence.

**Remark:** Note that this bound was first derived by Zhao et al. (2019). The first term measures the disagreement between the source and target labeling functions, the second term is the source error, and the third term quantifies the discrepancy between the marginal feature distributions. The bound identifies three key factors for successful task adaptation: small source risk, close marginal distributions, and consistent labeling functions across tasks.

**Further Analysis.** Lemma C.9 provides an upper bound showing that the generated classifier achieves a lower error than task-isolated classifiers when tasks share structure. This justifies using a hypernetwork instead of maintaining separate heads for each task: the shared layer effectively captures useful regularities while the frozen layer constrains deviation.

**Lemma C.10.** Let $\tilde{\mathbb{P}}_{\mathcal{S}}$ and $\tilde{\mathbb{P}}_{\mathcal{T}}$ denote the induced distribution over the feature space for each distribution $\mathbb{P}_{\mathcal{S}}$ and $\mathbb{P}_{\mathcal{T}}$ over the original input space. Then for any $h^{\mathcal{S}} \in \mathcal{H}, h^{\mathcal{T}} \in \mathcal{H}$, we have

$$|\epsilon^{\mathcal{S}}(h^{\mathcal{S}}, h^{\mathcal{T}}) - \epsilon^{\mathcal{T}}(h^{\mathcal{S}}, h^{\mathcal{T}})| \leq d_{\mathcal{H}}(\tilde{\mathbb{P}}_{\mathcal{S}}, \tilde{\mathbb{P}}_{\mathcal{T}}). \tag{C.31}$$

*Proof.* By definition, for any $h^{\mathcal{S}} \in \mathcal{H}, h^{\mathcal{T}} \in \mathcal{H}$, we have:

$$
\begin{aligned}
&\left|\epsilon^{\mathcal{S}}(h^{\mathcal{S}}, h^{\mathcal{T}}) - \epsilon^{\mathcal{T}}(h^{\mathcal{S}}, h^{\mathcal{T}})\right| \\
&= \sup_{h \in \mathcal{H}} \left|\mathbb{E}_{X \sim \mathbb{P}_{\mathcal{S}}}|h^{\mathcal{S}}(X) - h^{\mathcal{T}}(X)| - \mathbb{E}_{X \sim \mathbb{P}_{\mathcal{T}}}|h^{\mathcal{S}}(X) - h^{\mathcal{T}}(X)|\right| \\
&= \sup_{h \in \mathcal{H}} \left|\mathbb{E}_{X \sim \mathbb{P}_{\mathcal{S}}}[h^{\mathcal{S}}(X) - h^{\mathcal{T}}(X)] - \mathbb{E}_{X \sim \mathbb{P}_{\mathcal{T}}}[h^{\mathcal{S}}(X) - h^{\mathcal{T}}(X)]\right|.
\end{aligned}
\tag{C.32}
$$

Since $\| \cdot \|_\infty \leq 1$ for all $h \in \mathcal{H}$, we have $0 \leq |h^\mathcal{S}(X) - h^\mathcal{T}(X)| \leq 1$ for all $X \in \mathbb{P}_\mathcal{S}/\mathbb{P}_\mathcal{T}$, where $h^\mathcal{S}, h^\mathcal{T} \in \mathcal{H}$. Here we define a hypothesis space $\hat{\mathcal{H}} := \left\{ \mathrm{sgn}(|h^\mathcal{S}(X) - h^\mathcal{T}(X)| - z) \mid h^\mathcal{S}, h^\mathcal{T} \in \mathcal{H}, 0 \leq z \leq 1 \right\}$. Then we use Fubini's theorem to bound:

$$
\begin{aligned}
&\left| \mathbb{E}_{X \sim \mathbb{P}_\mathcal{S}}[|h^\mathcal{S}(X) - h^\mathcal{T}(X)|] - \mathbb{E}_{X \sim \mathbb{P}_\mathcal{T}}[|h^\mathcal{S}(X) - h^\mathcal{T}(X)|] \right| \\
&= \left| \int_0^1 (\Pr{}^\mathcal{S}(|h^\mathcal{S} - h^\mathcal{T}| > z) - \Pr{}^\mathcal{T}(|h^\mathcal{S} - h^\mathcal{T}| > z)) dz \right| \\
&\leq \int_0^1 \left| \Pr{}^\mathcal{S}(|h^\mathcal{S} - h^\mathcal{T}| > z) - \Pr{}^\mathcal{T}(|h^\mathcal{S} - h^\mathcal{T}| > z) \right| dz \\
&= \sup_{z \in [0,1]} \left| \Pr{}^\mathcal{S}(|h^\mathcal{S} - h^\mathcal{T}| > z) - \Pr{}^\mathcal{T}(|h^\mathcal{S} - h^\mathcal{T}| > z) \right|.
\end{aligned}
\tag{C.33}
$$

Combining Eq. (C.32) and Eq. (C.33), and in view of the definition of $\mathcal{H}$-divergence, we have:

$$
\begin{aligned}
&\sup_{h^\mathcal{S}, h^\mathcal{T} \in \mathcal{H}} \sup_{z \in [0,1]} \left| \Pr{}^\mathcal{S}(|h^\mathcal{S} - h^\mathcal{T}| > z) - \Pr{}^\mathcal{T}(|h^\mathcal{S} - h^\mathcal{T}| > z) \right| \\
&= 2 \sup_{\hat{h} \in \hat{\mathcal{H}}} \left| \Pr{}^\mathcal{S}(\hat{h}(X) = 1) - \Pr{}^\mathcal{T}(\hat{h}(X) = 1) \right| \\
&= 2 \sup_{\hat{h} \in \hat{\mathcal{H}}} \left| \Pr{}^\mathcal{S}(\hat{h}) - \Pr{}^\mathcal{T}(\hat{h}) \right| \\
&= d_{\hat{\mathcal{H}}}(\tilde{\mathbb{P}}_\mathcal{S}, \tilde{\mathbb{P}}_\mathcal{T}) \\
&\leq d_\mathcal{H}(\tilde{\mathbb{P}}_\mathcal{S}, \tilde{\mathbb{P}}_\mathcal{T}).
\end{aligned}
\tag{C.34}
$$

$\square$

**Theorem C.11** (Error Upper Bound with Task-specific Prototypes). Let $\tilde{\mathbb{P}}_\mathcal{S}$ and $\tilde{\mathbb{P}}_\mathcal{T}$ denote the induced distribution over the feature space for each distribution $\mathbb{P}_\mathcal{S}$ and $\mathbb{P}_\mathcal{T}$ over the original input space. The risk $\epsilon^\mathcal{T}(\hat{h})$ with task-specifc prototypes (with mixed weight $w_0 + w_1 = 1$, $w_0 \geq 0$, $w_1 \geq 0$, and $\hat{h} = w_0 \hat{h}^\mathcal{S} + w_1 \hat{h}^\mathcal{T}$) is upper bounded by:

$$
\epsilon^\mathcal{T}(\hat{h}) \leq \epsilon^\mathcal{T}(\hat{h}, h^\mathcal{S}) + \epsilon^\mathcal{T}(h^\mathcal{S}, h^\mathcal{T}).
\tag{C.35}
$$

*Proof.* The proof of the above theorem is as follows:

$$
\begin{aligned}
\epsilon^\mathcal{T}(\hat{h}) &= \epsilon^\mathcal{T}(\hat{h}, h^\mathcal{T}) \\
&= \mathbb{E}_{x \sim \mathbb{P}_\mathcal{T}}[|\hat{h}(x) - h^\mathcal{T}(x)|] \\
&= \mathbb{E}_{x \sim \mathbb{P}_\mathcal{T}}[|\hat{h}(x) - h^\mathcal{S}(x) + h^\mathcal{S}(x) - h^\mathcal{T}(x)|] \\
&\leq \mathbb{E}_{x \sim \mathbb{P}_\mathcal{T}}[|\hat{h}(x) - h^\mathcal{S}(x)|] + \mathbb{E}_{x \sim \mathbb{P}_\mathcal{T}}[|h^\mathcal{S}(x) - h^\mathcal{T}(x)|] \\
&= \epsilon^\mathcal{T}(\hat{h}, h^\mathcal{S}) + \epsilon^\mathcal{T}(h^\mathcal{S}, h^\mathcal{T})
\end{aligned}
\tag{C.36}
$$

**Comparison of Error Upper Bounds.** From Lemma C.10, our bound in Eq. (C.35) is further bounded by:

$$
\begin{aligned}
\epsilon^\mathcal{T}(\hat{h}) &\leq \epsilon^\mathcal{T}(\hat{h}, h^\mathcal{S}) + \epsilon^\mathcal{T}(h^\mathcal{S}, h^\mathcal{T}) \\
&\leq \epsilon^\mathcal{S}(\hat{h}, h^\mathcal{S}) + d_\mathcal{H}(\tilde{\mathbb{P}}_\mathcal{S}, \tilde{\mathbb{P}}_\mathcal{T}) + \epsilon^\mathcal{T}(h^\mathcal{S}, h^\mathcal{T}) \\
&= \epsilon^\mathcal{S}(\hat{h}) + \epsilon^\mathcal{T}(h^\mathcal{S}, h^\mathcal{T}) + d_\mathcal{H}(\tilde{\mathbb{P}}_\mathcal{S}, \tilde{\mathbb{P}}_\mathcal{T})
\end{aligned}
\tag{C.37}
$$

We further introduce a density ratio $\frac{\Pr^\mathcal{T}(x)}{\Pr^\mathcal{S}(x)}$ to represent the divergence between the source and target tasks:

$$\epsilon^{\mathcal{T}}(\hat{h}) = \int |\hat{h}(x) - h^{\mathcal{S}}(x)| \cdot \mathrm{Pr}^{\mathcal{T}}(x)dx + \epsilon^{\mathcal{T}}(h^{\mathcal{S}}, h^{\mathcal{T}})$$

$$= \int |\hat{h}(x) - h^{\mathcal{S}}(x)| \cdot \frac{\mathrm{Pr}^{\mathcal{T}}(x)}{\mathrm{Pr}^{\mathcal{S}}(x)} \cdot \mathrm{Pr}^{\mathcal{S}}(x)dx + \epsilon^{\mathcal{T}}(h^{\mathcal{S}}, h^{\mathcal{T}})$$

$$= \mathbb{E}_{x \sim \mathbb{P}_{\mathcal{S}}} \frac{\mathrm{Pr}^{\mathcal{T}}(x)}{\mathrm{Pr}^{\mathcal{S}}(x)} \cdot |\hat{h}(x) - h^{\mathcal{S}}(x)| + \epsilon^{\mathcal{T}}(h^{\mathcal{S}}, h^{\mathcal{T}}) \qquad \text{(C.38)}$$

$$= \frac{\mathrm{Pr}^{\mathcal{T}}(x)}{\mathrm{Pr}^{\mathcal{S}}(x)} \epsilon^{\mathcal{S}}(\hat{h}) + \epsilon^{\mathcal{T}}(h^{\mathcal{S}}, h^{\mathcal{T}})$$

$$= \epsilon^{\mathcal{S}}(\hat{h}) + \epsilon^{\mathcal{T}}(h^{\mathcal{S}}, h^{\mathcal{T}}),$$

Since the density ratio $\frac{\mathrm{Pr}^{\mathcal{T}}(x)}{\mathrm{Pr}^{\mathcal{S}}(x)}$ is generally unobservable, it is commonly treated as a constant and omitted. Next, we consider two cases. If $\epsilon^{\mathcal{S}}(h^{\mathcal{S}}, h^{\mathcal{T}}) \leq \epsilon^{\mathcal{T}}(h^{\mathcal{S}}, h^{\mathcal{T}})$, then we have:

$$\epsilon_0^{\mathcal{T}}(\hat{h}) \leq \epsilon^{\mathcal{S}}(h^{\mathcal{S}}, h^{\mathcal{T}}) + \epsilon^{\mathcal{S}}(\hat{h}) + d_{\mathcal{H}}(\tilde{\mathbb{P}}_{\mathcal{S}}, \tilde{\mathbb{P}}_{\mathcal{T}}), \qquad \text{(C.39)}$$

Given that

$$\epsilon^{\mathcal{T}}(\hat{h}) \leq \epsilon^{\mathcal{S}}(\hat{h}) + \epsilon^{\mathcal{T}}(h^{\mathcal{S}}, h^{\mathcal{T}})$$
$$\leq \epsilon^{\mathcal{S}}(h^{\mathcal{S}}, h^{\mathcal{T}}) + \epsilon^{\mathcal{S}}(\hat{h}) + d_{\mathcal{H}}(\tilde{\mathbb{P}}_{\mathcal{S}}, \tilde{\mathbb{P}}_{\mathcal{T}}), \qquad \text{(C.40)}$$

it follows that the error upper bound of $\epsilon^{\mathcal{T}}(\hat{h})$ is tighter than that of $\epsilon_0^{\mathcal{T}}(\hat{h})$.

Similarly, if $\epsilon^{\mathcal{S}}(h^{\mathcal{S}}, h^{\mathcal{T}}) \geq \epsilon^{\mathcal{T}}(h^{\mathcal{S}}, h^{\mathcal{T}})$, then:

$$\epsilon_0^{\mathcal{T}}(\hat{h}) \leq \epsilon^{\mathcal{T}}(h^{\mathcal{S}}, h^{\mathcal{T}}) + \epsilon^{\mathcal{S}}(\hat{h}) + d_{\mathcal{H}}(\tilde{\mathbb{P}}_{\mathcal{S}}, \tilde{\mathbb{P}}_{\mathcal{T}})$$
$$\leq \epsilon^{\mathcal{S}}(h^{\mathcal{S}}, h^{\mathcal{T}}) + \epsilon^{\mathcal{S}}(\hat{h}) + d_{\mathcal{H}}(\tilde{\mathbb{P}}_{\mathcal{S}}, \tilde{\mathbb{P}}_{\mathcal{T}}), \qquad \text{(C.41)}$$

which again implies that the error upper bound of $\epsilon^{\mathcal{T}}(\hat{h})$ is tighter than that of $\epsilon_0^{\mathcal{T}}(\hat{h})$.

$\square$

**Theorem C.12** (Error Upper Bound with Continual Cross-Task Knowledge Transfer). Let $\tilde{\mathbb{P}}_{G_{t-1}}$ and $\tilde{\mathbb{P}}_{G^t}$ denote the induced feature distributions of two consecutive target graphs at steps $t-1$ and $t$. The risk $\epsilon_t(\hat{h})$ with state-evolving prototypes (with mixed weight $w_{t-1} + w_t = 1$, $w_{t-1}, w_t \geq 0$, and $\hat{h} = w_{t-1}\hat{h}_{t-1} + w_t\hat{h}_t$) is upper bounded by:

$$\epsilon_t(\hat{h}) \leq \epsilon_t(\hat{h}, h_{t-1}) + \epsilon_t(h_{t-1}, h_t). \qquad \text{(C.42)}$$

*Proof.* Similarly, from Theorem C.11, we have:

$$\epsilon^{\mathcal{T}}(\hat{h}) \leq \epsilon^{\mathcal{T}}(\hat{h}, h^{\mathcal{S}}) + \epsilon^{\mathcal{T}}(h^{\mathcal{S}}, h^{\mathcal{T}}). \qquad \text{(C.43)}$$

By substituting $\mathcal{T}$ with $t$ and $\mathcal{S}$ with $t-1$, we obtain the following inequality, which holds throughout the continual task adaptation process:

$$\epsilon^t(\hat{h}) \leq \epsilon^t(\hat{h}, h^{t-1}) + \epsilon^t(h^{t-1}, h^t). \qquad \text{(C.44)}$$

That is,

$$\epsilon_t(\hat{h}) \leq \epsilon_t(\hat{h}, h_{t-1}) + \epsilon_t(h_{t-1}, h_t). \qquad \text{(C.45)}$$

**Comparison of Error Upper Bounds.** In particular, we can regard Theorem C.12 as a special case of Theorem C.11, where the $t$-th task serves as the initial target task for adaptation (i.e., modeling the task-specific prototype and transferring from the source task to the target task). Therefore, the same ordering of error upper bounds holds, and the bound remains tighter than that of $\epsilon_0^{\mathcal{T}}(\hat{h})$. $\square$

Table D.1: Key statistics of the graph datasets.

| Datasets | CoraFull | Arxiv | Reddit | Products |
|---|---|---|---|---|
| #Nodes | 19,793 | 169,343 | 227,853 | 2,449,028 |
| #Edges | 130,622 | 1,166,243 | 114,615,892 | 61,859,036 |
| #Classes | 70 | 40 | 40 | 46 |
| #Tasks | 35 | 20 | 20 | 23 |
| Avg. nodes per task | 660 | 8,467 | 11,393 | 122,451 |
| Avg. edges per task | 4,354 | 58,312 | 5,730,794 | 2,689,523 |

# D EXPERIMENT

## D.1 DATASET DESCRIPTION

Following CGLB (Zhang et al., 2022a), four large GCIL datasets are used in our experiments. The statistics of the used graph datasets are summarized in Table D.1.

- **CoraFull**[D.1]: It is a citation network containing 70 classes, where nodes represent papers and edges represent citation links between papers.

- **Arxiv**[D.2]: It is also a citation network between all Computer Science (CS) ARXIV papers indexed by MAG (Sinha et al., 2015). Each node in Arxiv denotes a CS paper and the edge between nodes represents a citation between them. The nodes are classified into 40 subject areas. The node features are computed as the average word-embedding of all words in the title and abstract.

- **Reddit**[D.3]: It encompasses Reddit posts generated in September 2014, with each post classified into distinct communities or subreddits. Specifically, nodes represent individual posts, and the edges between posts exist if a user has commented on both posts. Node features are derived from various attributes, including post title, content, comments, post score, and the number of comments.

- **Products**[D.4]: It is an Amazon product co-purchasing network, where nodes represent products sold in Amazon and the edges between nodes indicate that the products are purchased together. The node features are constructed with the dimensionality-reduced bag-of-words of the product descriptions.

## D.2 BASELINE DETAILS

- **EWC** (Kirkpatrick et al., 2017) is a regularization-based method that adds a quadratic penalty on the model parameters according to their importance to the previous tasks to maintain the performance on previous tasks.

- **MAS** (Aljundi et al., 2018) preserves the parameters important to previous tasks based on the sensitivity of the predictions to the changes in the parameters.

- **GEM** (Lopez-Paz & Ranzato, 2017) stores representative data in the episodic memory and proposes to modify the gradients of the current task with the gradient calculated on the memory data to tackle the forgetting problem.

- **LwF** (Li & Hoiem, 2017) employs knowledge distillation to minimize the discrepancy between the logits of the old and the new models to preserve the knowledge of the previous tasks.

- **TWP** (Liu et al., 2021) proposes to preserve the important parameters in the topological aggregation and loss minimization for previous tasks via regularization terms.

- **ERGNN** (Zhou & Cao, 2021) is a replay-based method that constructs memory data by storing representative nodes selected from previous tasks.

---

[D.1] https://docs.dgl.ai/en/1.1.x/generated/dgl.data.CoraFullDataset.html
[D.2] https://ogb.stanford.edu/docs/nodeprop/#ogbn-arxiv
[D.3] https://docs.dgl.ai/en/1.1.x/generated/dgl.data.RedditDataset.html#dgl.data.RedditDataset
[D.4] https://ogb.stanford.edu/docs/nodeprop/#ogbn-products

- **SSM** (Zhang et al., 2022c) incorporates the explicit topological information of selected nodes in the form of sparsified computation subgraphs into the memory for graph continual learning.

- **SEM** (Zhang et al., 2023c) improves SSM by storing the most informative topological information via the Ricci curvature-based graph sparsification technique.

- **CaT** (Liu et al., 2023) condenses each graph to a small synthesized replayed graph and stores it in a condensed graph memory with historical replay graphs. Moreover, graph continual learning is accomplished by updating the model directly with the condensed graph memory.

- **DeLoMe** (Niu et al., 2024a) learns lossless prototypical node representations as the memory to capture the holistic graph information of previous tasks. A debiased GCL loss function is further devised to address the data imbalance between the classes in the memory data and the current data.

- **TPP** (Niu et al., 2024b) transductively captures task-specific prototypes utilizing a Laplacian smoothing-based matching approach, achieving 100% task ID prediction accuracy and 0% forgetting ratio.

## D.3 CONFIGURATIONS

We conduct the experiments with:

- Operating System: Ubuntu 20.04 LTS.

- CPU: Intel(R) Xeon(R) Gold 6240 CPU @ 2.60GHz, 256GB RAM.

- GPU: Tesla V100 PCIe 32GB GPU.

- Software: Python 3.7, Pytorch 1.8, CUDA 11.0, and Pytorch-Geometric 2.0.1.

## D.4 COMPARISONS ON TRAINABLE PARAMETERS AND RUNTIME

Based on the analysis in Appendix B.2, the number of trainable parameters for TPP is $P_{\text{tpp}} = T(d+1)C$, while Graph2Hyper requires $P_{\text{hyper}} = (d+1)h + Td$, where $d$ is the feature dimension, $C$ is the number of classes per task, $T$ is the number of tasks, and $h$ is the hidden dimension of the hypernetwork (with $h \ll d$). Using the benchmark configurations and assuming ($C = 2$) for each task, we computed the **total trainable parameter counts** for Graph2Hyper and the TPP classifier on all datasets, as summarized in Table D.2. From these results, we observe that Graph2Hyper is more parameter-efficient than assigning a separate classifier head for each task.

Table D.2: Trainable parameter comparison between Graph2Hyper and TPP across datasets.

| Method | CoraFull (T=35) | Arxiv (T=20) | Reddit (T=20) | Products (T=23) |
|---|---|---|---|---|
| TPP | $2 \times 35(d+1) = 70d + 70$ | $2 \times 20(d+1) = 40d + 40$ | $2 \times 20(d+1) = 40d + 40$ | $2 \times 23(d+1) = 46d + 46$ |
| Graph2Hyper | $(d+1)h + 35d = 36d + h$ | $(d+1)h + 20d = 21d + h$ | $(d+1)h + 20d = 21d + h$ | $(d+1)h + 23d = 24d + h$ |

We also provide statistical comparisons of training time, inference time, and memory usage between Graph2Hyper and TPP across all datasets, as shown in Table D.3. Graph2Hyper uses substantially fewer trainable parameters than TPP, and the training time is comparable or shorter due to the shared hypernetwork. With the parameter-efficient design of Graph2Hyper, the training process also benefits from reduced computational overhead.

## D.5 ADDITIONAL RESULTS ON BROADER CLASS COUNTS PER TASK

Since the default GCIL setting fixes the number of classes per task to two, we extend the evaluation to verify the scalability of our method under broader configurations. Specifically, we conduct additional experiments with larger task sizes. From Table D.4 and Table D.5, we observe that Graph2Hyper maintains consistent performance across all four datasets, even when each task contains 3 or 5 classes. Extending the setting to uneven class counts or imbalanced label distributions introduces additional difficulty, and we view this as an important direction for future investigation.

Table D.3: Comparison on runtime and peak GPU memory usage of Graph2Hyper and TPP.

| Method | CoraFull | Arxiv | Reddit | Products |
|---|---|---|---|---|
| **TPP** | | | | |
| Training Time (s) | 294.42 | 148.34 | 347.83 | 1475.75 |
| Inference Time (s) | 51.36 | 12.28 | 58.07 | 134.54 |
| Max GPU Mem. (MiB) | 1636.65 | 270.10 | 8344.95 | 11780.42 |
| **Graph2Hyper** | | | | |
| Training Time (s) | 44.76 | 22.96 | 124.36 | 430.63 |
| Inference Time (s) | 11.62 | 2.20 | 38.53 | 110.71 |
| Max GPU Mem. (MiB) | 383.87 | 214.72 | 8323.13 | 11766.61 |

Table D.4: Performance (% ± standard deviation) under the GCIL setting (3 classes per task).

| Method | CoraFull | | Arxiv | | Reddit | | Products | |
|---|---|---|---|---|---|---|---|---|
| | AA/% ↑ | AF/% ↑ | AA/% ↑ | AF/% ↑ | AA/% ↑ | AF/% ↑ | AA/% ↑ | AF/% ↑ |
| DMSG | 76.08±0.08 | 2.06±0.54 | 48.33±0.54 | 2.24±0.77 | 84.53±4.59 | -14.81±4.69 | 66.00±0.40 | -0.9±1.6 |
| TPP | 94.82±0.64 | 0.0±0.0 | 83.23±0.62 | 0.0±0.0 | 97.90±0.74 | 0.0±0.0 | 88.30±0.55 | 0.0±0.0 |
| **Graph2Hyper** | 97.10±0.31 | 0.0±0.0 | 84.17±0.38 | 0.0±0.0 | 98.57±0.63 | 0.0±0.0 | 89.27±0.31 | 0.0±0.0 |

Table D.5: Performance (% ± standard deviation) under the GCIL setting (5 classes per task).

| Method | CoraFull | | Arxiv | | Reddit | | Products | |
|---|---|---|---|---|---|---|---|---|
| | AA/% ↑ | AF/% ↑ | AA/% ↑ | AF/% ↑ | AA/% ↑ | AF/% ↑ | AA/% ↑ | AF/% ↑ |
| DMSG | 76.95±0.17 | 3.58±0.41 | 45.55±0.54 | 4.95±0.59 | 82.51±8.59 | -15.05±9.52 | 64.23±1.10 | -4.20±0.50 |
| TPP | 95.21±0.82 | 0.0±0.0 | 74.97±0.41 | 0.0±0.0 | 91.79±0.80 | 0.0±0.0 | 83.39±0.47 | 0.0±0.0 |
| **Graph2Hyper** | 96.17±0.64 | 0.0±0.0 | 76.25±0.11 | 0.0±0.0 | 96.46±0.29 | 0.0±0.0 | 87.22±0.43 | 0.0±0.0 |

Table D.6: Graph2Hyper performance and task-ID prediction accuracy under different task orders.

| Task Orders | CoraFull | | Arxiv | | Reddit | | Products | |
|---|---|---|---|---|---|---|---|---|
| | AA/% | task-ID Acc/% | AA/% | task-ID Acc/% | AA/% | task-ID Acc/% | AA/% | task-ID Acc/% |
| Ascending | 94.83 | 100.00 | 87.18 | 100.00 | 99.48 | 100.00 | 94.71 | 100.00 |
| Descending | 94.73 | 100.00 | 86.33 | 100.00 | 99.43 | 100.00 | 94.32 | 100.00 |
| Random | 95.24 | 100.00 | 87.01 | 100.00 | 99.57 | 100.00 | 89.14 | 100.00 |

Table D.7: Graph2Hyper performance and task-ID prediction accuracy under different noise levels.

| Degree | CoraFull | | | Arxiv | | | Reddit | | | Products | | |
|---|---|---|---|---|---|---|---|---|---|---|---|---|
| | AA/% | AF/% | task-ID Acc/% | AA/% | AF/% | task-ID Acc/% | AA/% | AF/% | task-ID Acc/% | AA/% | AF/% | task-ID Acc/% |
| 0.01 | 94.20 | 0.00 | 100.00 | 85.89 | 0.00 | 100.00 | 97.46 | 0.00 | 100.00 | 93.97 | 0.00 | 100.00 |
| 0.03 | 94.06 | 0.00 | 100.00 | 84.86 | 0.00 | 100.00 | 96.78 | 0.00 | 100.00 | 93.51 | 0.00 | 100.00 |
| 0.05 | 93.64 | 0.00 | 100.00 | 84.27 | 0.00 | 100.00 | 95.89 | 0.00 | 100.00 | 92.47 | 0.00 | 100.00 |
| 0.1 | 93.39 | 0.00 | 100.00 | 83.86 | 0.00 | 100.00 | 95.31 | 0.00 | 100.00 | 92.28 | 0.00 | 100.00 |
| 0.3 | 93.15 | 0.00 | 100.00 | 83.35 | 0.00 | 100.00 | 94.13 | 0.00 | 100.00 | 92.14 | 0.00 | 100.00 |

## D.6 RESULTS OF TASK-ID ACCURACY ALONGSIDE AA/AF

We include results under different task orders that report task-ID prediction accuracy along with AA and AF in Table D.6. The results show that the ability to predict task IDs correctly does not depend on the sequence in which tasks appear.

## D.7 RESULTS WITH DIFFERENT GNN BACKBONES

We extended our experiments by incorporating GCN, GIN, and GAT as backbones, and we compared the performance of Graph2Hyper on all four datasets. The results are summarized in Table D.8.

**Analysis.** Under all these encoders, Graph2Hyper maintains similar performance. This is because the GNN encoder remains frozen throughout the entire process. The differences observed in the table

Table D.8: Performance of Graph2Hyper with different GNN backbones.

| Backbone | CoraFull | | Arxiv | | Reddit | | Products | |
|---|---|---|---|---|---|---|---|---|
| | AA/% | AF/% | AA/% | AF/% | AA/% | AF/% | AA/% | AF/% |
| GCN | 94.36±0.46 | 0.00±0.00 | 87.68±0.39 | 0.00±0.00 | 98.88±0.10 | 0.00±0.00 | 94.69±0.90 | 0.00±0.00 |
| GIN | 94.49±0.40 | 0.00±0.00 | 86.26±0.08 | 0.00±0.00 | 99.41±0.04 | 0.00±0.00 | 93.58±1.44 | 0.00±0.00 |
| GAT | 93.92±0.45 | 0.00±0.00 | 87.52±0.12 | 0.00±0.00 | 98.64±0.13 | 0.00±0.00 | 94.44±0.55 | 0.00±0.00 |
| SGC | 94.83±0.55 | 0.00±0.00 | 87.18±0.69 | 0.00±0.00 | 99.47±0.05 | 0.00±0.00 | 94.71±1.65 | 0.00±0.00 |

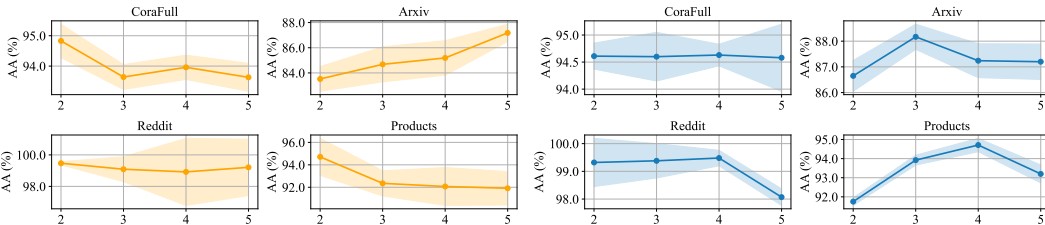

Figure D.1: The sensitivity of SGC layers.      Figure D.2: The sensitivity of parameter $K$.

primarily reflect the inherent performance variation of the encoders on the datasets. The components that influence GCIL performance are the classifier head and the hypernetwork.

### D.8 ROBUSTNESS OF TASK-ID PREDICTION UNDER RANDOM NOISE

We include additional experiments where each task's graph samples are perturbed with random noise to evaluate Graph2Hyper's performance and task-ID prediction accuracy in Table D.7. Specifically, we randomly add or remove edges from graphs while fully adhering to the GCIL setting. As the noise level increases, performance decreases moderately across the four datasets, which is expected. Importantly, task-ID prediction remains stable. Because the hypernetwork generates a dedicated classifier head for each task, correct task-ID estimation naturally supports forget-free behavior.

**Analysis of global testing.** Following the TPP setting, the current Graph2Hyper architecture does not support global testing where nodes from different tasks are mixed. This is a limitation. Under the current setup, only our method and TPP achieve both forget-free and rehearsal-free learning. A potential future direction is to design a hypernetwork that can increase the output dimension of the classifier head as the number of learned classes grows, enabling global mixed-task evaluation while maintaining forget-free behavior.

### D.9 ADDITIONAL RESULTS OF PARAMETER STUDY

**Sensitivity Analysis of SGC Layers.** To study the sensitivity of the number of layers in the SGC backbone of Graph2Hyper, we vary it within $\{2, 3, 4, 5\}$. The AA corresponding to different choices is reported in Figure D.1. The results indicate that Graph2Hyper is robust to reasonable selections of SGC layers, while careful tuning is still important for achieving optimal performance.

**Sensitivity Analysis of Order $K$.** To study the sensitivity of $K$-th order to smooth the node features, we add sensitivity sweeps over the propagation order $K$, and the results are shown in Figure D.2. Results demonstrate that the performance is sensitive to changes and contains a reasonable range across different datasets.

## E USAGE STATEMENT OF LARGE LANGUAGE MODELS (LLMS)

In the preparation of this paper, large language models (LLMs) were used solely as general-purpose writing assistants. Specifically, we employed an LLM for grammar correction and language polishing to improve readability and presentation.

