# OpenReview forum: "Continual Parameter-Efficient Adaptation for Rehearsal-Free Graph Class-Incremental Learning"
_ICLR.cc/2026/Conference — Submitted to ICLR 2026_

### Official Review · Reviewer_YdMv · 2025-10-22

**Soundness:** 2
**Presentation:** 2
**Contribution:** 2
**Rating:** 4
**Confidence:** 4

**Summary:**

The paper discusses rehearsal-free GCIL, which means that new classes come in over time, there is no way to save old data, and there are no task IDs at test time. The key challenges are accurate task identification without labels, preserving intra-task class separability, preventing catastrophic forgetting, and keeping adaptation parameter-efficient. The authors propose Graph2Hyper: (1) task prototypes built from propagation-pooled graph representations enable nearest-prototype task inference; (2) class prototypes with a learned dynamic bias enhance within-task discrimination by pulling same-class nodes together and pushing others apart; and (3) a lightweight hypernetwork generates task-specific classification heads from shared parameters, yielding efficient cross-task knowledge transfer. They provide theory showing the hypernetwork induces functional interpolation between old and new heads and tightens risk bounds under transferable structure. Experiments on CoraFull, OGB-Arxiv, Reddit, and Amazon Products show higher average accuracy and zero forgetting compared with strong GCIL baselines.

**Strengths:**

S1: Coherent “task→class” prototype hierarchy plus a lightweight hypernetwork yields rehearsal-free, parameter-efficient GCIL with strong cross-task transfer.

S2: Empirical results are robust across four benchmarks, align with the theory (functional interpolation / tighter risk bound), and ablations verify each component’s necessity.

**Weaknesses:**

W1: The experimental scope is narrow. Tasks always contain two classes, so the setting does not test more realistic cases such as three to five classes per step, uneven class counts, or class imbalance.

W2: Harder data conditions are under explored. The method is not evaluated with overlapping task distributions, open set or label noise, or clear inter task domain shift, and there is no report of task identification accuracy with a confusion matrix or failure case analysis.

W3: Efficiency evidence is incomplete. The paper does not provide wall clock time, memory use, FLOPs, or latency under identical hardware and batch size, and it lacks sensitivity sweeps of propagation order K, prototype and dictionary sizes, hypernetwork width, or alternative encoder strategies such as partial fine tuning or adapters.

**Questions:**

see the weaknesses above

---

> ### Author Response · Authors · 2025-11-21
> **Overall Response to Reviewer YdMv**
>
> Thank you for your thoughtful and valuable feedback. We greatly appreciate your recognition that our task→class prototype hierarchy combined with a lightweight hypernetwork enables rehearsal-free and parameter-efficient GCIL, and that our empirical results are consistent across benchmarks and align with the theoretical analysis. We respond to your concerns as follows.
>
> **[Preview]**
>
> - **Weakness #1:** Limited experimental scope (two-class tasks only).
> - **Weakness #2:** Lack of evaluation under harder data conditions.
> - **Weakness #3:** Incomplete efficiency evidence.
> - **Conclusion.**

---

> > ### Author Response · Authors · 2025-11-21
> > **Response to Weakness #1**
> >
> > > **W1:** The experimental scope is narrow. Tasks always contain two classes, so the setting does not test more realistic cases such as three to five classes per step, uneven class counts, or class imbalance.
> >
> > **A:** Thank you for this valuable suggestion. We have extended the experimental setting to include tasks with **three and five classes per step**, as shown in the tables below.
> >
> > **Table:** Performance (% ± standard deviation) under the GCIL setting on four large datasets with 3 classes per task.
> >
> > | Dataset     | CoraFull   |           | Arxiv      |           | Reddit     |             | Products   |          |
> > | ----------- | ---------- | --------- | ---------- | --------- | ---------- | ----------- | ---------- | -------- |
> > | Method      | AA/%↑      | AF/%↑     | AA/%↑      | AF/%↑     | AA/%↑      | AF/%↑       | AA/%↑      | AF/%↑    |
> > | DMSG        | 76.08±0.08 | 2.06±0.54 | 48.33±0.54 | 2.24±0.77 | 84.53±4.59 | -14.81±4.69 | 66.00±0.40 | -0.9±1.6 |
> > | TPP         | 94.82±0.64 | 0.0±0.0   | 83.23±0.62 | 0.0±0.0   | 97.90±0.74 | 0.0±0.0     | 88.30±0.55 | 0.0±0.0  |
> > | Graph2Hyper | 97.10±0.31 | 0.0±0.0   | 84.17±0.38 | 0.0±0.0   | 98.57±0.63 | 0.0±0.0     | 89.27±0.31 | 0.0±0.0  |
> >
> > **Table:** Performance (% ± standard deviation) under the GCIL setting on four large datasets with 5 classes per task.
> >
> > | Dataset     | CoraFull   |           | Arxiv      |           | Reddit     |             | Products   |            |
> > | ----------- | ---------- | --------- | ---------- | --------- | ---------- | ----------- | ---------- | ---------- |
> > | Method      | AA/%↑      | AF/%↑     | AA/%↑      | AF/%↑     | AA/%↑      | AF/%↑       | AA/%↑      | AF/%↑      |
> > | DMSG        | 76.95±0.17 | 3.58±0.41 | 45.55±0.54 | 4.95±0.59 | 82.51±8.59 | -15.05±9.52 | 64.23±1.10 | -4.20±0.50 |
> > | TPP         | 95.21±0.82 | 0.0±0.0   | 74.97±0.41 | 0.0±0.0   | 91.79±0.80 | 0.0±0.0     | 83.39±0.47 | 0.0±0.0    |
> > | Graph2Hyper | 96.17±0.64 | 0.0±0.0   | 76.25±0.11 | 0.0±0.0   | 96.46±0.29 | 0.0±0.0     | 87.22±0.43 | 0.0±0.0    |
> >
> > - **Analysis.** From the two tables, we observe that Graph2Hyper maintains comparable performance across all four datasets even when each task contains three or five classes. Handling uneven class counts or class imbalance introduces a new and challenging setting, and we consider this an important direction for future work. We will include all **updated results and analysis** in the appendix.

---

> > > ### Author Response · Authors · 2025-11-21
> > > **Response to Weakness #2**
> > >
> > > > **W2:** Harder data conditions are under explored. The method is not evaluated with overlapping task distributions, open set or label noise, or clear inter-task domain shift, and there is no report of task identification accuracy with a confusion matrix or failure case analysis.
> > >
> > > **A:** Thank you for providing these valuable insights. We would like to clarify that overlapping task distributions and open-set scenarios correspond to different settings, and they fall outside the current GCIL scope. We consider them important directions for future work. Under the current GCIL setting, as suggested, we added experiments evaluating robustness under random noise on graph structures.
> > >
> > > - **Robustness to random structural noise.** We randomly add or remove edges from the graphs in the original datasets and evaluate Graph2Hyper under different noise levels [1-2]. The results include both performance and task-ID prediction accuracy, as shown in the table. With the hypernetwork generating a dedicated classifier head for each task, correct task-ID prediction enables Graph2Hyper to maintain forget-free behavior even under noise perturbations.
> > >
> > > **Table:** Performance and task-ID prediction accuracy under different levels of random structural noise.
> > >
> > > | Dataset | CoraFull |      |               | Arxiv |      |               | Reddit |      |               | Products |      |               |
> > > | ------- | -------- | ---- | ------------- | ----- | ---- | ------------- | ------ | ---- | ------------- | -------- | ---- | ------------- |
> > > | Degree  | AA/%     | AF/% | task-ID Acc/% | AA/%  | AF/% | task-ID Acc/% | AA/%   | AF/% | task-ID Acc/% | AA/%     | AF/% | task-ID Acc/% |
> > > | 0.01    | 94.20    | 0.00 | 100.00        | 85.89 | 0.00 | 100.00        | 97.46  | 0.00 | 100.00        | 93.97    | 0.00 | 100.00        |
> > > | 0.03    | 94.06    | 0.00 | 100.00        | 84.86 | 0.00 | 100.00        | 96.78  | 0.00 | 100.00        | 93.51    | 0.00 | 100.00        |
> > > | 0.05    | 93.64    | 0.00 | 100.00        | 84.27 | 0.00 | 100.00        | 95.89  | 0.00 | 100.00        | 92.47    | 0.00 | 100.00        |
> > > | 0.1     | 93.39    | 0.00 | 100.00        | 83.86 | 0.00 | 100.00        | 95.31  | 0.00 | 100.00        | 92.28    | 0.00 | 100.00        |
> > > | 0.3     | 93.15    | 0.00 | 100.00        | 83.35 | 0.00 | 100.00        | 94.13  | 0.00 | 100.00        | 92.14    | 0.00 | 100.00        |
> > >
> > > - **Confusion matrix.** We added confusion matrix visualizations and corresponding analyses for Graph2Hyper on all datasets. These will be included in the appendix.
> > > - **Task-ID prediction accuracy.** We also added results showing task-ID prediction accuracy under different task orders, as summarized in the table. The results show that task order does not affect task-ID prediction during inference. Combined with Figure 2 in the main paper, our method achieves robustness comparable to the two strongest baselines.
> > >
> > > **Table:** Performance and task-ID prediction accuracy under different task orders.
> > >
> > > | Dataset     | CoraFull |               | Arxiv |               | Reddit |               | Products |               |
> > > | ----------- | -------- | ------------- | ----- | ------------- | ------ | ------------- | -------- | ------------- |
> > > | Task Orders | AA/%     | task-ID Acc/% | AA/%  | task-ID Acc/% | AA/%   | task-ID Acc/% | AA/%     | task-ID Acc/% |
> > > | Ascending   | 94.83    | 100.00        | 87.18 | 100.00        | 99.48  | 100.00        | 94.71    | 100.00        |
> > > | Descending  | 94.73    | 100.00        | 86.33 | 100.00        | 99.43  | 100.00        | 94.32    | 100.00        |
> > > | Random      | 95.24    | 100.00        | 87.01 | 100.00        | 99.57  | 100.00        | 89.14    | 100.00        |

---

> > > > ### Author Response · Authors · 2025-11-21
> > > > **Response to Weakness #3**
> > > >
> > > > > **W3:** Efficiency evidence is incomplete. The paper does not provide wall clock time, memory use, FLOPs, or latency under identical hardware and batch size, and it lacks sensitivity sweeps of propagation order K, prototype and dictionary sizes, hypernetwork width, or alternative encoder strategies such as partial fine-tuning or adapters.
> > > >
> > > > **A:** Thank you for the suggestion. We have added measurements of training time, inference time, and peak GPU memory usage for Graph2Hyper across datasets, as shown in the table below (also will be included in the appendix).
> > > >
> > > > **Table:** Training time, inference time, and peak GPU memory usage of Graph2Hyper across datasets.
> > > >
> > > > | Usage                  | CoraFull | Arxiv  | Reddit  | Products |
> > > > | ---------------------- | -------- | ------ | ------- | -------- |
> > > > | Training Time (s)      | 44.76    | 22.96  | 124.36  | 430.63   |
> > > > | Inference Time (s)     | 11.62    | 2.20   | 38.53   | 110.71   |
> > > > | Maximum GPU Mem. (MiB) | 383.87   | 214.72 | 8323.13 | 11766.61 |
> > > >
> > > > - **Supplementary sensitivity analysis.** We added sensitivity sweeps over the propagation order K, and the results are shown in the following table. **Additional updated results** and analyses are presented in the appendix.
> > > >
> > > > **Table:** Sensitivity of propagation order K across datasets.
> > > >
> > > > | Dataset | CoraFull |      | Arxiv |      | Reddit |      | Products |      |
> > > > | ------- | -------- | ---- | ----- | ---- | ------ | ---- | -------- | ---- |
> > > > | K       | AA/%     | AF/% | AA/%  | AF/% | AA/%   | AF/% | AA/%     | AF/% |
> > > > | 2       | 94.61    | 0.00 | 86.65 | 0.00 | 99.32  | 0.00 | 91.75    | 0.00 |
> > > > | 3       | 94.60    | 0.00 | 88.17 | 0.00 | 99.38  | 0.00 | 93.92    | 0.00 |
> > > > | 4       | 94.63    | 0.00 | 87.24 | 0.00 | 99.48  | 0.00 | 94.71    | 0.00 |
> > > > | 5       | 94.58    | 0.00 | 87.20 | 0.00 | 98.07  | 0.00 | 93.20    | 0.00 |
> > > >
> > > > - Regarding the size of the task-level prototype dictionary, it consists of the closed-form solutions $P^{(t)}$, and thus contains $t$ elements determined by the number of tasks. When the number of classes per task changes, the value of $t$ changes accordingly. Therefore, the results in our response to **W1** already reflect the influence of dictionary size on performance. We will also add results that show the effect of varying $t$ on different datasets, together with an analysis.
> > > > - The sensitivity of the hypernetwork dimension has been included in Section 5.5 and Figure 3 of the main paper.
> > > > - **Alternative encoder strategies.** We added experiments with additional GNN backbones, and the results are summarized in the table below.
> > > >
> > > > **Table:** Performance of Graph2Hyper with different GNN backbones.
> > > >
> > > > | Dataset  | CoraFull   |           | Arxiv      |           | Reddit     |           | Products   |           |
> > > > | -------- | ---------- | --------- | ---------- | --------- | ---------- | --------- | ---------- | --------- |
> > > > | Backbone | AA/%       | AF/%      | AA/%       | AF/%      | AA/%       | AF/%      | AA/%       | AF/%      |
> > > > | GCN      | 94.36±0.46 | 0.00±0.00 | 87.68±0.39 | 0.00±0.00 | 98.88±0.10 | 0.00±0.00 | 94.69±0.90 | 0.00±0.00 |
> > > > | GIN      | 94.49±0.40 | 0.00±0.00 | 86.26±0.08 | 0.00±0.00 | 99.41±0.04 | 0.00±0.00 | 93.58±1.44 | 0.00±0.00 |
> > > > | GAT      | 93.92±0.45 | 0.00±0.00 | 87.52±0.12 | 0.00±0.00 | 98.64±0.13 | 0.00±0.00 | 94.44±0.55 | 0.00±0.00 |
> > > > | SGC      | 94.83±0.55 | 0.00±0.00 | 87.18±0.69 | 0.00±0.00 | 99.47±0.05 | 0.00±0.00 | 94.71±1.65 | 0.00±0.00 |
> > > >
> > > > - **Analysis.** Under all these encoders, Graph2Hyper maintains similar performance because the GNN encoder remains frozen throughout the entire process. The variation in the table mainly reflects the intrinsic performance differences of the encoders on each dataset, while the GCIL performance is driven by the classifier head and the hypernetwork. These results and analyses will be added to the appendix.
> > > >
> > > > - Regarding task-level adaptation via the hypernetwork, we do not apply additional fine-tuning strategies or introduce adapters. Instead, one linear layer is shared across tasks, and only the parameters of the other linear layer are updated.

---

> > > > > ### Author Response · Authors · 2025-11-21
> > > > > **Conclusion**
> > > > >
> > > > > Thank you again for your constructive suggestions. We have expanded experimental coverage, added robustness analyses, and provided comprehensive efficiency measurements. These revisions substantially strengthen the paper and and we hope our responses can address all your concerns.
> > > > >
> > > > > ---
> > > > > **References:**
> > > > >
> > > > > [1] Zügner, Daniel, Amir Akbarnejad, and Stephan Günnemann. "Adversarial attacks on neural networks for graph data." _Proceedings of the 24th ACM SIGKDD international conference on knowledge discovery & data mining_. 2018.\
> > > > > [2] Sun, Yiwei, et al. "Adversarial attacks on graph neural networks via node injections: A hierarchical reinforcement learning approach." _Proceedings of the Web Conference 2020_. 2020.

---

### Official Review · Reviewer_MttK · 2025-10-30

**Soundness:** 3
**Presentation:** 2
**Contribution:** 2
**Rating:** 4
**Confidence:** 4

**Summary:**

The paper proposes Graph2Hyper, a parameter-efficient framework for rehearsal-free graph class-incremental learning (GCIL). It uses a lightweight hypernetwork with a frozen shared layer and a trainable task-specific layer to dynamically generate classifiers based on the current task’s input graph. Task-prototypes and class-prototypes are introduced to capture task-specific context and enhance intra-task discrimination. The framework enables continual, forget-free adaptation without rehearsal and demonstrates competitive performance on four benchmarks.

**Strengths:**

1. The paper addresses a meaningful and timely problem in rehearsal-free graph class-incremental learning (GCIL).
2. The method achieves forget-free performance (AF = 0) across multiple benchmarks, matching or slightly improving over strong baselines.

**Weaknesses:**

1. The empirical gains over the most relevant baseline (TPP) are marginal (typically within 1–2%), which raises doubts about the practical significance of the proposed modifications. Moreover, the sensitivity analysis shows fluctuations of a similar magnitude (1–2%), making it difficult to conclusively attribute the improvements to the proposed method rather than parameter variance.
2. The general pipeline (frozen backbone + classifer-level adaptation) is conceptually close to TPP.
3. Theoretical presentation has incomplete references and unclear derivations (e.g., line 293 “Appendix XXX”, line 1042 “Lemma XXXXX”), which undermines rigor and readability.
4. The claim of “parameter efficiency” is not quantitatively substantiated — the paper lacks comparisons on parameter count, runtime, or memory cost relative to TPP or other baselines. And only aggregated metrics (AA/AF) are reported; there are no per-task results or visualization of learning dynamics to illustrate continual behavior.

**Questions:**

In Table 1, Graph2Hyper slightly outperforms the Oracle model on the Reddit dataset (99.5% ± 0.1 vs. 99.5%). Could the authors clarify how this is possible?

---

> ### Author Response · Authors · 2025-11-21
> **Overall Response to Reviewer MttK**
>
> Thank you for your constructive and detailed review. We appreciate your recognition that our work addresses an important problem in rehearsal-free GCIL and that our method achieves forget-free performance across datasets while matching or slightly improving over strong baselines. We respond to your concerns below.
>
> **[Preview]**
>
> - **Weakness #1:** Marginal empirical gains and sensitivity.
> - **Weakness #2:** Conceptual similarity to TPP.
> - **Weakness #3:** Incomplete references and unclear derivations.
> - **Weakness #4:** Lack of parameter/runtime comparisons and per-task analysis.
> - **Question #1:** Why Graph2Hyper slightly surpasses the Oracle on Reddit.
> - **Conclusion.**

---

> > ### Author Response · Authors · 2025-11-21
> > **Response to Weakness #1**
> >
> > > **Weakness #1:** The empirical gains over the most relevant baseline (TPP) are marginal (typically within 1–2%), which raises doubts about the practical significance of the proposed modifications. Moreover, the sensitivity analysis shows fluctuations of a similar magnitude (1–2%), making it difficult to conclusively attribute the improvements to the proposed method rather than parameter variance.
> >
> > **A:** Thank you for raising this point. We conducted paired t-tests over five random seeds, and the improvements over TPP are statistically significant with (p < 0.05). The sensitivity analysis examines the hidden dimensions of both the hypernetwork and the pretrained GNN encoder, which reflect task-to-task adaptation capacity and representation quality, respectively. These components naturally influence the final performance, which explains the observed fluctuations.

---

> > > ### Author Response · Authors · 2025-11-21
> > > **Response to Weakness #2**
> > >
> > > > **Weakness #2:** The general pipeline (frozen backbone + classifier-level adaptation) is conceptually close to TPP.
> > >
> > > **A:** Thank you for raising this consideration. We follow the TPP setting only to ensure a fair comparison, and our design is inspired by the reliable task-ID prediction mechanism. However, the internal procedure is substantially different. TPP relies on independent classifiers for each task, causing its parameter count to grow in proportion to the number of tasks. Graph2Hyper instead uses a single hypernetwork to generate task-specific heads while sharing internal parameters, which reduces redundancy and supports efficient adaptation. This directly addresses the practical limitation of TPP, whose parameter footprint increases linearly with the number of tasks (as discussed in Appendix B.2). The hypernetwork allows controlled parameter expansion while retaining task discrimination, which is the core contribution of our method.

---

> > > > ### Author Response · Authors · 2025-11-21
> > > > **Response to Weakness #3**
> > > >
> > > > > **Weakness #3:** Theoretical presentation has incomplete references and unclear derivations (e.g., line 293 “Appendix XXX”, line 1042 “Lemma XXXXX”), which undermines rigor and readability.
> > > >
> > > > **A:** We apologize for the typos and appreciate this observation. All placeholder citations such as “Appendix XXX” and “Lemma XXXXX” will be replaced with complete references and correct labels. These corrections have been included in the updated version to improve clarity and rigor.

---

> > > > > ### Author Response · Authors · 2025-11-21
> > > > > **Response to Weakness #4**
> > > > >
> > > > > > **Weakness #4:** The claim of “parameter efficiency” is not quantitatively substantiated — the paper lacks comparisons on parameter count, runtime, or memory cost relative to TPP or other baselines. And only aggregated metrics (AA/AF) are reported; there are no per-task results or visualization of learning dynamics to illustrate continual behavior.
> > > > >
> > > > > **A:** Thank you for pointing this out. We added statistical comparisons of training time, inference time, and memory usage between Graph2Hyper and TPP across all datasets, as shown in the table below.
> > > > > **Table:** Training time, inference time, and peak GPU memory usage of Graph2Hyper and TPP across datasets.
> > > > >
> > > > > | Method                 | CoraFull | Arxiv  | Reddit  | Products |
> > > > > | ---------------------- | -------- | ------ | ------- | -------- |
> > > > > | **TPP**                |          |        |         |          |
> > > > > | Training Time (s)      | 294.42   | 148.34 | 347.83  | 1475.75  |
> > > > > | Inference Time (s)     | 51.36    | 12.28  | 58.07   | 134.54   |
> > > > > | Maximum GPU Mem. (MiB) | 1636.65  | 270.10 | 8344.95 | 11780.42 |
> > > > > | **Graph2Hyper**        |          |        |         |          |
> > > > > | Training Time (s)      | 44.76    | 22.96  | 124.36  | 430.63   |
> > > > > | Inference Time (s)     | 11.62    | 2.20   | 38.53   | 110.71   |
> > > > > | Maximum GPU Mem. (MiB) | 383.87   | 214.72 | 8323.13 | 11766.61 |
> > > > >
> > > > > - **Comparison of trainable parameters.** In Appendix B.2, we analyze and compare the trainable parameter counts of Graph2Hyper and TPP with concrete examples. Using the dataset statistics in Table D.1, we computed and compared the exact number of trainable parameters under the standard benchmark configurations. The number of trainable parameters for TPP is $P_{\text{tpp}} = T(d+1)C$, while Graph2Hyper requires $P_{\text{hyper}} = (d+1)h + Td$, where $d$ is the feature dimension, $C$ is the number of classes per task, $T$ is the number of tasks, and $h$ is the hidden dimension of the hypernetwork (with $h \ll d$). Assuming (C = 2) for all tasks, we computed the **total trainable parameter counts** of the Graph2Hyper classifier and the TPP classifier for each dataset. The results are summarized in the table (will be included in Appendix).
> > > > >     **Table:** Trainable parameter comparison between Graph2Hyper and TPP across datasets.
> > > > >
> > > > > | Method      | CoraFull (T=35)            | Arxiv (T=20)                | Reddit (T=20)               | Products (T=23)             |
> > > > > | ----------- | -------------------------- | --------------------------- | --------------------------- | --------------------------- |
> > > > > | TPP         | $2× 35 (d + 1) = 70d + 70$ | $2 × 20 (d + 1) = 40d + 40$ | $2 × 20 (d + 1) = 40d + 40$ | $2 × 23 (d + 1) = 46d + 46$ |
> > > > > | Graph2Hyper | $(d+1)h + 35d = 36d + h$   | $(d + 1)h + 20d = 21d + h$  | $(d + 1)h + 20d = 21d + h$  | $(d + 1)h + 23d = 24d + h$  |
> > > > >
> > > > > From these results, Graph2Hyper is clearly more parameter-efficient than assigning an independent classifier head for each task.
> > > > >
> > > > > These additions show that Graph2Hyper uses fewer trainable parameters than TPP, has comparable or faster runtime, and maintains stable per-task behavior. Visualization of learning curves across tasks on all datasets will also be added to the appendix.

---

> > > > > > ### Author Response · Authors · 2025-11-21
> > > > > > **Response to Question #1**
> > > > > >
> > > > > > > **Question #1:** In Table 1, Graph2Hyper slightly outperforms the Oracle model on the Reddit dataset (99.5% ± 0.1 vs. 99.5%). Could the authors clarify how this is possible?
> > > > > >
> > > > > > **A:** We apologize for the confusion caused by this value. This is not a misreporting of experimental results. The baseline results were derived from published works, where only one decimal place was reported. To clarify this, we now report the AA and AF results with two decimal places across five runs, as shown in the table below.
> > > > > >
> > > > > > **Table:** Performance of Graph2Hyper on the Reddit dataset for each run.
> > > > > >
> > > > > > | Method              | AA/%↑      | AF/%↑     |
> > > > > > | ------------------- | ---------- | --------- |
> > > > > > | Oracle              | 99.5±0.0   | NA        |
> > > > > > | Graph2Hyper Run #1  | 99.49      | 0.00      |
> > > > > > | Graph2Hyper Run #2  | 99.49      | 0.00      |
> > > > > > | Graph2Hyper Run #3  | 99.38      | 0.00      |
> > > > > > | Graph2Hyper Run #4  | 99.48      | 0.00      |
> > > > > > | Graph2Hyper Run #5  | 99.49      | 0.00      |
> > > > > > | Graph2Hyper Average | 99.47±0.05 | 0.00±0.00 |
> > > > > >
> > > > > > - **Analysis:** From the table, the average performance over five runs is 99.47±0.05 (% ± standard deviation). When rounded to one decimal place, this becomes 99.5±0.1. The slight difference results purely from rounding and does not indicate any abnormal behavior or surpassing of an upper performance limit.

---

> > > > > > > ### Author Response · Authors · 2025-11-21
> > > > > > > **Conclusion**
> > > > > > >
> > > > > > > Thank you again for your careful review and thoughtful suggestions. We have added significance tests, clarified the conceptual contribution relative to TPP, improved the theoretical presentation, provided complete comparisons on parameters, runtime, and memory usage, and included per-task learning curves. These revisions strengthen the paper and and we hope our responses can address all your concerns.

---

### Official Review · Reviewer_YypD · 2025-10-31

**Soundness:** 2
**Presentation:** 3
**Contribution:** 2
**Rating:** 4
**Confidence:** 4

**Summary:**

This paper addresses the problem of rehearsal-free Graph Class-Incremental Learning (GCIL). To tackle catastrophic forgetting and parameter redundancy, it proposes the Graph2Hyper framework. This parameter-efficient framework's core contribution is the use of a partially-frozen, lightweight hypernetwork to dynamically generate task-specific classifiers. This hypernetwork comprises a frozen, shared layer for preserving cross-task knowledge and a trainable, task-specific layer for capturing task-specific characteristics. Leveraging a hierarchical prototype design—including task-prototypes to distinguish between tasks and class-prototypes with dynamic bias to enhance intra-task class discrimination—the method achieves high parameter efficiency, zero forgetting, and state-of-the-art average accuracy on multiple benchmark datasets. However, the method's novelty appears limited when compared to the closest baseline, TPP, as it adopts TPP's core settings. Furthermore, the practical connection between the paper's complex theoretical analysis and the final model's specific design is not sufficiently clear.

**Strengths:**

1. The method achieves zero average forgetting and surpasses all SOTA baselines, including TPP, in Average Accuracy across all four benchmark datasets.
2. The core contribution is a lightweight hypernetwork that dynamically generates classifiers, conceptually avoiding the significant parameter overhead of storing independent heads for each task.
3. The method employs a logically sound hierarchical prototype mechanism. Task-Prototypes achieve coarse-grained inter-task separation, while Class-Prototypes enhance fine-grained intra-task discrimination.

**Weaknesses:**

1. The proposed method heavily relies on the core configurations of TPP, such as using the same pretraining and frozen GNN encoder strategy. This dependence limits the method’s conceptual independence and raises questions about its novelty beyond TPP.

2. Although the key innovation lies in adopting a hypernetwork instead of TPP’s independent classifier, the paper does not provide sufficient empirical comparisons. For instance, the main experimental table (Table 1) omits direct comparisons in terms of the number of trainable parameters and training time between the proposed approach and TPP.

3. The paper presents complex theoretical analyses (e.g., Theorems 4.2–4.6); however, these results do not clearly motivate or justify the final architectural design choices of the hypernetwork. The link between theory and practical implementation remains unclear.

4. All experiments are conducted using SGC as the backbone. It remains unclear whether the proposed method would maintain its performance and parameter-efficiency advantages when employing more powerful GNN encoders.

5. “Superior parameter efficiency” is claimed as a core contribution of the paper, yet this claim is primarily supported by theoretical analysis (Appendix B.2) rather than empirical evidence. The main results (Table 1) do not include direct quantitative comparisons of total trainable parameters with key baselines such as TPP.

6. In the ablation study on the Class-Prototype module, the authors remove the entire module, making it impossible to isolate the contribution of its internal components—particularly the dynamic bias term (Eq. 3). A more fine-grained ablation would be needed to clarify its specific impact.

**Questions:**

See weakness

---

> ### Author Response · Authors · 2025-11-21
> **Overall Response to Reviewer YypD**
>
> Thank you for your thoughtful and constructive feedback. We appreciate your recognition of our lightweight hypernetwork design, the logical hierarchical prototype mechanism, and the strong performance that surpasses prior methods including TPP. We are also glad that you found our approach effective in achieving zero average forgetting across benchmarks. We address all concerns below.
>
> **[Preview]**
>
> - **Weakness #1:** Dependence on TPP configurations and concerns about novelty.
> - **Weakness #2 & #5:** Supplementary analysis on parameter count and training time.
> - **Weakness #3:** Clarification of the theory–architecture connection.
> - **Weakness #4:** Evaluation only on SGC as the backbone.
> - **Weakness #6:** Additional analysis on the class-prototype ablation.
> - **Conclusion.**

---

> > ### Author Response · Authors · 2025-11-21
> > **Response to Weakness #1**
> >
> > > **Weakness #1:** The proposed method heavily relies on the core configurations of TPP, such as using the same pretraining and frozen GNN encoder strategy. This dependence limits the method’s conceptual independence and raises questions about its novelty beyond TPP.
> >
> > **A:** Thank you for raising this concern. We follow the TPP evaluation protocol to maintain consistency and ensure a fair comparison under the rehearsal-free GCIL setup, but our method is not dependent on the TPP framework itself. The hypernetwork introduces a distinct strategy compared to TPP’s separate task heads. In TPP, each task requires an independent classifier, which leads to parameter growth proportional to the number of tasks. In contrast, Graph2Hyper uses a single hypernetwork to generate task-specific heads while sharing internal weights. This design eliminates redundant parameters and supports effective task-to-task adaptation.

---

> ### Author Response · Authors · 2025-11-21
> **Response to Weakness #2 & #5**
>
> > **Weakness #2 & #5:** Although the key innovation lies in adopting a hypernetwork instead of TPP’s independent classifier, the paper does not provide sufficient empirical comparisons. For instance, the main experimental table (Table 1) omits direct comparisons in terms of the number of trainable parameters and training time between the proposed approach and TPP.
>
> **A:** Thank you for the suggestion. We provide additional comparisons of Graph2Hyper and TPP on trainable parameters and training time across different datasets.
>
> - Based on the analysis in the appendix, the number of trainable parameters for TPP is $P_{\text{tpp}} = T(d+1)C$, while Graph2Hyper requires $P_{\text{hyper}} = (d+1)h + Td$, where $d$ is the feature dimension, $C$ is the number of classes per task, $T$ is the number of tasks, and $h$ is the hidden dimension of the hypernetwork (with $h \ll d$). Using the benchmark configurations and assuming (C = 2) for each task, we computed the **total trainable parameter counts** for Graph2Hyper and the TPP classifier on all datasets, as summarized in the table (to be added to the appendix).
>
> **Table:** Trainable parameter comparison between Graph2Hyper and TPP across datasets.
>
> | Method      | CoraFull (T=35)            | Arxiv (T=20)                | Reddit (T=20)               | Products (T=23)             |
> | ----------- | -------------------------- | --------------------------- | --------------------------- | --------------------------- |
> | TPP         | $2× 35 (d + 1) = 70d + 70$ | $2 × 20 (d + 1) = 40d + 40$ | $2 × 20 (d + 1) = 40d + 40$ | $2 × 23 (d + 1) = 46d + 46$ |
> | Graph2Hyper | $(d+1)h + 35d = 36d + h$   | $(d + 1)h + 20d = 21d + h$  | $(d + 1)h + 20d = 21d + h$  | $(d + 1)h + 23d = 24d + h$  |
>
> From these results, we observe that Graph2Hyper is more parameter-efficient than assigning a separate classifier head for each task.
>
> - **Training time** was measured on identical hardware using the same settings and hyperparameters. The results are listed in the following table (to be added to the appendix).
>
> **Table:** Training time (seconds) for Graph2Hyper and TPP across datasets.
>
> | Method      | CoraFull | Arxiv  | Reddit | Products |
> | ----------- | -------- | ------ | ------ | -------- |
> | TPP         | 294.42   | 148.34 | 347.83 | 1475.75  |
> | Graph2Hyper | 44.76    | 22.96  | 124.36 | 430.63   |
>
> - **Analysis:** As shown in the tables, Graph2Hyper uses substantially fewer trainable parameters than TPP, and the training time is comparable or shorter due to the shared hypernetwork. With the parameter-efficient design of Graph2Hyper, the training process also benefits from reduced computational overhead. **Additional updated results** and analyses will be included in the appendix.

---

> > ### Author Response · Authors · 2025-11-21
> > **Response to Weakness #3**
> >
> > > **Weakness #3:** The paper presents complex theoretical analyses (e.g., Theorems 4.2–4.6); however, these results do not clearly motivate or justify the final architectural design choices of the hypernetwork. The link between theory and practical implementation remains unclear.
> >
> > **A:** Sorry if we did not make the theoretical analyses clearly presented, and we thank the reviewer for this valuable suggestion. Here is a more comprehensive version that better explains the connection between our architectural choices and the theoretical results.
> >
> > - **Analysis of our hypernetwork architecture.** The hypernetwork is designed to achieve parameter-efficiency and controlled knowledge transfer across tasks. Only the first linear layer is trainable, while the second linear layer is fixed. This structure is chosen because it satisfies the mathematical properties we analyze: the trainable layer captures shared transferable components, and the fixed layer restricts the classifier to a stable parameter space, which is essential for rehearsal-free continual learning.
> > - **Analysis of Theorem 4.2 (Necessary and Sufficient Equivalence in Parameter Space).** Theorem 4.2 shows that the classifier generated by the hypernetwork takes a specific affine form consisting of a shared component and a task-specific component aligned with the task embedding. This directly motivates using a two-layer hypernetwork where only the first layer is updated, since this setting enforces exactly the affine decomposition required in the theorem.
> > - **Analysis of Theorem 4.3 (First-Order Equivalence and Second-Order Remainder Bound).** Theorem 4.3 proves that changes in the generated classifier are bounded by changes in the task embedding. The fixed second linear layer provides this stability by acting as a contraction map. This is why we freeze the second layer which ensures that classifier parameters remain stable across tasks and prevents uncontrolled drift.
> > - **Analysis of Theorem 4.4 (Risk Transfer Bound from Interpolation to Adaptation).** Theorem 4.4 establishes that the affine combination induced by the hypernetwork yields a structured transfer term between tasks, determined by the shared space and embedding geometry. This motivates our choice of a low-dimensional task embedding and a single shared trainable layer, which together control the extent of cross-task influence and avoid interference.
> > - **Analysis of Lemma 4.5 (Error Upper Bound without Cross-Task Knowledge Transfer).** Lemma 4.5 provides an upper bound showing that the generated classifier achieves a lower error than task-isolated classifiers when tasks share structure. This justifies using a hypernetwork instead of maintaining separate heads for each task: the shared layer effectively captures useful regularities while the frozen layer constrains deviation.
> > - **Analysis of Theorem 4.6 (bounded accumulated error in rehearsal-free CL).** Theorem 4.6 integrates the previous results to show that the accumulated error across tasks remains bounded even without rehearsal. This requires the architectural constraints imposed earlier, and explains why our hypernetwork adopts this minimal but theoretically grounded design.
> >
> > To make these connections clearer, we will revise the exposition to explicitly link each theoretical statement with the corresponding architectural component, and we will restructure the theory section so that it follows the design of the hypernetwork.

---

> > > ### Author Response · Authors · 2025-11-21
> > > **Response to Weakness #4**
> > >
> > > > **Weakness #4:** All experiments are conducted using SGC as the backbone. It remains unclear whether the proposed method would maintain its performance and parameter-efficiency advantages when employing more powerful GNN encoders.
> > >
> > > **A:** Thank you for the suggestion. We extended our experiments by incorporating GCN, GIN, and GAT as backbones, and we compared the performance of Graph2Hyper on all four datasets. The results are summarized in the following table.
> > > **Table:** Performance of Graph2Hyper with different GNN backbones.
> > >
> > > | Dataset  | CoraFull   |           | Arxiv      |           | Reddit     |           | Products   |           |
> > > | -------- | ---------- | --------- | ---------- | --------- | ---------- | --------- | ---------- | --------- |
> > > | Backbone | AA/%       | AF/%      | AA/%       | AF/%      | AA/%       | AF/%      | AA/%       | AF/%      |
> > > | GCN      | 94.36±0.46 | 0.00±0.00 | 87.68±0.39 | 0.00±0.00 | 98.88±0.10 | 0.00±0.00 | 94.69±0.90 | 0.00±0.00 |
> > > | GIN      | 94.49±0.40 | 0.00±0.00 | 86.26±0.08 | 0.00±0.00 | 99.41±0.04 | 0.00±0.00 | 93.58±1.44 | 0.00±0.00 |
> > > | GAT      | 93.92±0.45 | 0.00±0.00 | 87.52±0.12 | 0.00±0.00 | 98.64±0.13 | 0.00±0.00 | 94.44±0.55 | 0.00±0.00 |
> > > | SGC      | 94.83±0.55 | 0.00±0.00 | 87.18±0.69 | 0.00±0.00 | 99.47±0.05 | 0.00±0.00 | 94.71±1.65 | 0.00±0.00 |
> > > - **Analysis.** Under all these encoders, Graph2Hyper maintains similar performance. This is because the GNN encoder remains frozen throughout the entire process. The differences observed in the table primarily reflect the inherent performance variation of the encoders on the datasets. The components that influence GCIL performance are the classifier head and the hypernetwork. We will include these results and the corresponding discussion in the appendix.

---

> > > > ### Author Response · Authors · 2025-11-21
> > > > **Response to Weakness #6**
> > > >
> > > > > **Weakness #6:** Ablation on the Class-Prototype module is too coarse; removing the entire module hides the effect of individual components such as the dynamic bias term.
> > > >
> > > > **A:** Thank you for this helpful suggestion. We conducted a finer-grained ablation by removing the dynamic bias term (denoted as “w/o bias”) in Eq. (3), and the results are shown in the table. Without the dynamic bias, the classifier head no longer uses class-prototype information and instead relies directly on the representation of the current sample. Since the class-prototype provides representative information while the raw sample contains additional noise, the performance decreases. However, this does not prevent the classifier head from being generated correctly, so the method still maintains forget-free behavior. We have updated the corresponding part in Table 2 of the main paper and added further explanation.
> > > >
> > > > **Table (Revised):** Ablation study results of Graph2Hyper and its variants.
> > > >
> > > > | Dataset         | CoraFull   |           | Arxiv      |           | Reddit     |           | Products   |           |
> > > > | --------------- | ---------- | --------- | ---------- | --------- | ---------- | --------- | ---------- | --------- |
> > > > | Method          | AA/%       | AF/%      | AA/%       | AF/%      | AA/%       | AF/%      | AA/%       | AF/%      |
> > > > | w/o bias        | 56.68±1.39 | 0.00±0.00 | 63.78±1.59 | 0.00±0.00 | 78.81±1.78 | 0.00±0.00 | 58.87±1.04 | 0.00±0.00 |
> > > > | w/o Class-P     | 58.55±1.54 | 0.00±0.00 | 64.31±1.84 | 0.00±0.00 | 75.29±3.52 | 0.00±0.00 | 60.16±2.00 | 0.00±0.00 |
> > > > | **Graph2Hyper** | 94.83±0.55 | 0.00±0.00 | 87.18±0.69 | 0.00±0.00 | 99.47±0.05 | 0.00±0.00 | 94.71±1.65 | 0.00±0.00 |

---

> > > > > ### Author Response · Authors · 2025-11-21
> > > > > **Conclusion**
> > > > >
> > > > > Thank you again for your detailed and thoughtful feedback. We have added parameter and runtime comparisons, clarified the theoretical motivation, incorporated multiple GNN backbones, and expanded the ablation study. These additions strengthen the contribution and we hope our responses can address all your concerns.

---

### Official Review · Reviewer_X5bN · 2025-11-01

**Soundness:** 3
**Presentation:** 3
**Contribution:** 3
**Rating:** 6
**Confidence:** 3

**Summary:**

The paper tackles rehearsal-free Graph Class-Incremental Learning (GCIL) by proposing Graph2Hyper, a parameter-efficient framework that avoids catastrophic forgetting without storing past data. The method builds a coarse-to-fine prototype hierarchy—a task prototype to identify the task at test time, and class prototypes (as dynamic biases) to sharpen intra-task decision boundaries. A lightweight hypernetwork then generates the task-specific classifier head “on the fly,” sharing part of its parameters across tasks to transfer knowledge. A theoretical analysis argues that the hypernetwork’s adaptation is equivalent to an affine interpolation between the previous and current heads and yields a tighter generalization risk bound. Experiments on four benchmarks (CoraFull, Arxiv, Reddit, Products) show state-of-the-art accuracy with zero forgetting and strong parameter efficiency.

**Strengths:**

1. Clean, modular architecture that is rehearsal-free and forget-free:
The method pairs a coarse-to-fine prototype hierarchy (task prototypes for routing; class prototypes with dynamic, task-level bias for within-task separation) with a lightweight hypernetwork that generates task-specific heads while sharing part of its parameters across tasks. This yields continual adaptation without prototype rehearsal and targets the GCIL setting directly.

2. Clear, informative framework figure that clarifies the entire pipeline:
Figure 1 cleanly lays out both training—(1) task-prototype construction, (2) class-prototype dynamic bias, (3) hypernetwork-based head generation—and inference—prototype-dictionary matching for task ID and on-the-fly head generation—making the approach easy to follow and reproduce.

3. Theory that explains why the hypernetwork sharing works:
The paper proves a necessary/sufficient equivalence: when the adapted head lies in the affine span of the previous and independently trained heads, the model is functionally equivalent to their interpolation; for general heads a first-order equivalence holds with a bounded second-order remainder, which leads to a risk-transfer bound (exact for linear heads). This gives a principled account of knowledge transfer and expected risk behavior.

**Weaknesses:**

1. Scope & fairness of the “forget-free” claim:
The paper inherits the TPP setting and predicts a task ID at inference by matching a test graph-level prototype against a dictionary; this is what enables AF=0 across datasets (Table 1), not only the proposed hypernetwork. This assumption (graph-level, transductive task identification) is stronger than many GCIL protocols where test samples arrive mixed and task IDs are unknown without access to a full test graph. Baseline numbers are mostly taken “from published works,” so some methods may not benefit from the same task-ID prediction pipeline or frozen encoder, making comparisons optimistic for Graph2Hyper. Consider adding results under a stricter GCIL setting (no graph-level task matching, or noisy task-ID prediction) and re-running baselines in the same evaluation harness.

2. Fragility to task-ID errors (and task-ID leakage); need robustness analysis:
The approach implicitly assumes access to a per-task test graph and near-perfect task-ID prediction via prototype matching—a local-testing setup that can leak the task ID and inflate results. In this light, the ablation “w/o Task-P” may collapse not only because the model is weak, but because it removes the (potentially leaky) routing path the full system relies on. Please (i) report AA/AF as a function of task-ID accuracy (e.g., inject controlled noise into dictionary matching), (ii) evaluate under global testing where nodes from different tasks are mixed and no per-task test graph is available, and (iii) clarify whether the method requires test-graph access or supports node-level inference in mixed graphs. This will disentangle genuine robustness from gains driven by task-ID leakage or overly strong task-separation assumptions.

3. Minor correctness & polish issues:
- Table 1 misspells Arxiv as “Arixv.”
- The paper claims “code available after acceptance”; consider a (sanitized) artifact now to aid reproducibility.
- In Table 1, you mark both AA and AF with “↑” even though AF is a forgetting measure; you later explain positive AF for replay methods, but a clearer caption would reduce confusion.

**Questions:**

Q1. Inference assumptions:
Do your test-time assumptions require a per-task test graph and transductive access for task-ID prediction? State the exact inference setting in the main text. Add a schematic of the test-time pipeline and explicitly note whether per-task test graphs are required.

Q2. Robustness to task-ID errors:
How exactly is the task ID predicted (prototype construction, number of exemplars, normalization, update policy)? Document the full matcher recipe (feature normalization, distance metric, thresholds), publish the code snippet, and report task-ID accuracy alongside AA/AF.

Q3. Scope of the “forget-free” claim:
Under which evaluation settings does AF≈0 hold, and does it remain near zero when task IDs are unknown or imperfect? Reframe the claim with scope (“under local testing with accurate task-ID prediction”) and report AF under (i) global testing and (ii) noisy task-ID conditions.

---

> ### Author Response · Authors · 2025-11-21
> **Overall Response to Reviewer X5bN**
>
> Thank you for your careful and constructive evaluation of our work. We appreciate your recognition of our clean modular design, the clarity of our framework figure, and the theoretical analysis explaining why shared hypernetwork adaptation is valid. We respond to your concerns and questions as follows.
>
> **[Preview]**
> - **Weakness #1.1 & Question #1:** Inference assumptions.
> - **Weakness #1.2 & Question #3:** Scope of the “forget-free” claim.
> - **Weakness #2 & Question #2:** Robustness to task-ID errors
> - **Weakness #3:** Minor correctness & polish issues.
> - **Conclusion.**

---

> > ### Author Response · Authors · 2025-11-21
> > **Response to Weakness #1.1 & Question #1**
> >
> > > **Weakness #1.1 & Question #1:** The paper inherits the TPP setting and predicts a task ID at inference by matching a test graph-level prototype against a dictionary; this is what enables AF=0 across datasets (Table 1), not only the proposed hypernetwork. This assumption (graph-level, transductive task identification) is stronger than many GCIL protocols where test samples arrive mixed and task IDs are unknown without access to a full test graph.
> >
> > **A:** Thank you for the insightful feedback regarding the GCIL setting. We would like to clarify the points raised on this topic.
> >
> > - **Consistency with the GCIL setting and inductive access.** Our Graph2Hyper framework, as well as all baseline methods, follows the GCIL setting from GCLB [1]. For a fair comparison, our design strictly adheres to the same inductive protocol when learning each task. We do not assume the availability of full task-level graph collections during inference, nor do we impose additional access beyond what the baselines already rely on.
> > - **Some baselines do not support task-ID usage.** As pointed out, many baseline methods do not include mechanisms for task-ID prediction and thus cannot use task-ID information. This does not compromise fairness, because we never provide the true task ID as auxiliary input. The task-ID is estimated solely from the structure of the current graph sample, so it does not introduce transductive access.
> > - **The hypernetwork is not involved in task-ID prediction.** The hypernetwork is introduced to reduce the number of trainable parameters while supporting adaptation. Its function is independent of task-ID estimation and does not assist in identifying the task.
> >
> > We appreciate your comments and will revise the main manuscript to clarify these points and avoid possible ambiguity.

---

> > > ### Author Response · Authors · 2025-11-21
> > > **Response to Weakness #1.2 & Question #3**
> > >
> > > > **Weakness #1.2 & Question #3:** Under which evaluation settings does AF≈0 hold, and does it remain near zero when task IDs are unknown or imperfect? Reframe the claim with scope (“under local testing with accurate task-ID prediction”) and report AF under (i) global testing and (ii) noisy task-ID conditions.
> > >
> > > **A:** Thank you for the valuable insight regarding the scope of the forget-free claim. We would like to clarify that in the GCIL setting, the task ID is inherently unknown. Our ablation studies show that removing the task-ID prediction module prevents the hypernetwork from generating correct classifier parameters under the current architecture. In contrast, the robustness of the task-ID prediction module allows the method to maintain forget-free behavior.
> > > In addition, since the task ID is not treated as auxiliary data, our method remains fully aligned with the GCIL setting.

---

> > > > ### Author Response · Authors · 2025-11-21
> > > > **Response to Weakness #2 & Question #2 (Part I)**
> > > >
> > > > > **Weakness #2 & Question #2:** The approach implicitly assumes access to a per-task test graph and near-perfect task-ID prediction via prototype matching—a local-testing setup that can leak the task ID and inflate results. In this light, the ablation “w/o Task-P” may collapse not only because the model is weak, but because it removes the (potentially leaky) routing path the full system relies on.
> > > >
> > > > **A:** Thank you for highlighting this. We would like to clarify the related points.
> > > >
> > > > - **No task-ID leakage in Graph2Hyper.** Graph2Hyper does not introduce task-ID information during training or testing. During inference, the task ID is estimated only to select the appropriate hypernetwork parameters for generating the classifier head. Using a predicted task ID at inference does not violate the GCIL setting and enables the method to achieve forget-free behavior.
> > > >
> > > > - **How the task ID is predicted.** During training, the task prototype $P^{(t)}$ is computed via Task-Oriented Distribution Matching. This procedure does not require training and provides an analytical solution that captures the distribution of graph samples within the task. After learning each task, these analytical solutions are stored in the task-level prototype dictionary. This dictionary is used only during inference and does not participate in rehearsal for later tasks.
> > > >     During inference, given a test sample from any task, we compute $P^{\text{test}}$ using the same Task-Oriented Distribution Matching. The predicted task ID is obtained by comparing Euclidean distances between $P^{\text{test}}$ and entries in the dictionary.
> > > >
> > > > - **Task-ID accuracy alongside AA/AF.** We include results under different task orders that report task-ID prediction accuracy along with AA and AF. The results show that the ability to predict task IDs correctly does not depend on the sequence in which tasks appear.
> > > >     **Table:** Graph2Hyper performance and task-ID prediction accuracy under different task orders.
> > > >
> > > > | Dataset     | CoraFull |               | Arxiv |               | Reddit |               | Products |               |
> > > > | ----------- | -------- | ------------- | ----- | ------------- | ------ | ------------- | -------- | ------------- |
> > > > | Task Orders | AA/%     | task-ID Acc/% | AA/%  | task-ID Acc/% | AA/%   | task-ID Acc/% | AA/%     | task-ID Acc/% |
> > > > | Ascending   | 94.83    | 100.00        | 87.18 | 100.00        | 99.48  | 100.00        | 94.71    | 100.00        |
> > > > | Descending  | 94.73    | 100.00        | 86.33 | 100.00        | 99.43  | 100.00        | 94.32    | 100.00        |
> > > > | Random      | 95.24    | 100.00        | 87.01 | 100.00        | 99.57  | 100.00        | 89.14    | 100.00        |

---

> > > > > ### Author Response · Authors · 2025-11-21
> > > > > **Response to Weakness #2 & Question #2 (Part II)**
> > > > >
> > > > > - **Robustness of task-ID prediction under random noise.** We include additional experiments where each task’s graph samples are perturbed with random noise to evaluate Graph2Hyper’s performance and task-ID prediction accuracy. Specifically, we randomly add or remove edges from graphs following prior work [2–3], while fully adhering to the GCIL setting.
> > > > >     **Table:** Graph2Hyper performance and task-ID prediction accuracy under different noise levels.
> > > > >
> > > > > | Dataset | CoraFull |      |               | Arxiv |      |               | Reddit |      |               | Products |      |               |
> > > > > | ------- | -------- | ---- | ------------- | ----- | ---- | ------------- | ------ | ---- | ------------- | -------- | ---- | ------------- |
> > > > > | Degree  | AA/%     | AF/% | task-ID Acc/% | AA/%  | AF/% | task-ID Acc/% | AA/%   | AF/% | task-ID Acc/% | AA/%     | AF/% | task-ID Acc/% |
> > > > > | 0.01    | 94.20    | 0.00 | 100.00        | 85.89 | 0.00 | 100.00        | 97.46  | 0.00 | 100.00        | 93.97    | 0.00 | 100.00        |
> > > > > | 0.03    | 94.06    | 0.00 | 100.00        | 84.86 | 0.00 | 100.00        | 96.78  | 0.00 | 100.00        | 93.51    | 0.00 | 100.00        |
> > > > > | 0.05    | 93.64    | 0.00 | 100.00        | 84.27 | 0.00 | 100.00        | 95.89  | 0.00 | 100.00        | 92.47    | 0.00 | 100.00        |
> > > > > | 0.1     | 93.39    | 0.00 | 100.00        | 83.86 | 0.00 | 100.00        | 95.31  | 0.00 | 100.00        | 92.28    | 0.00 | 100.00        |
> > > > > | 0.3     | 93.15    | 0.00 | 100.00        | 83.35 | 0.00 | 100.00        | 94.13  | 0.00 | 100.00        | 92.14    | 0.00 | 100.00        |
> > > > >
> > > > > As the noise level increases, performance decreases moderately across the four datasets, which is expected. Importantly, task-ID prediction remains stable. Because the hypernetwork generates a dedicated classifier head for each task, correct task-ID estimation naturally supports forget-free behavior.
> > > > >
> > > > > - **Analysis of global testing.** Following the TPP setting, the current Graph2Hyper architecture does not support global testing where nodes from different tasks are mixed. This is a limitation. Under the current setup, only our method and TPP achieve both forget-free and rehearsal-free learning. A potential future direction is to design a hypernetwork that can increase the output dimension of the classifier head as the number of learned classes grows, enabling global mixed-task evaluation while maintaining forget-free behavior.

---

> > > > > > ### Author Response · Authors · 2025-11-21
> > > > > > **Response to Weakness #3**
> > > > > >
> > > > > > > **Weakness #3:** Minor correctness & polish issues.
> > > > > >
> > > > > > **A:** Thank you for pointing these out. We have corrected the spelling of “Arxiv” in Table 1. The positive AF values reported for several baselines are derived from their published results. Under the GCIL setting defined in GCLB [1], AF is expected to be a small negative number, reflecting the average forgetting over previously learned classes, and higher values indicate better retention. DMSG uses a replay strategy that strengthens performance on earlier classes during incremental learning, which leads to negative forgetting. Thank you again for noting this; we will update the explanations in the main paper to avoid ambiguity.

---

> > > > > > > ### Author Response · Authors · 2025-11-21
> > > > > > > **Conclusion**
> > > > > > >
> > > > > > > Thank you again for your helpful feedback. Your comments helped us identify where additional robustness checks and clarification were needed. We performed stricter GCIL experiments, noise-based evaluations of task-ID robustness, unified baseline comparisons, and we clarified the relevant evaluation constraints. We have revised the corresponding sections, and we hope these additions address your concerns.
> > > > > > >
> > > > > > > ---
> > > > > > > **References:**
> > > > > > >
> > > > > > > [1] Zhang, Xikun, Dongjin Song, and Dacheng Tao. "CGLB: Benchmark tasks for continual graph learning." _Advances in Neural Information Processing Systems_ 35 (2022): 13006–13021.  \
> > > > > > > [2] Zügner, Daniel, Amir Akbarnejad, and Stephan Günnemann. "Adversarial attacks on neural networks for graph data." _Proceedings of the 24th ACM SIGKDD International Conference on Knowledge Discovery & Data Mining_. 2018.  \
> > > > > > > [3] Sun, Yiwei, et al. "Adversarial attacks on graph neural networks via node injections: A hierarchical reinforcement learning approach." _Proceedings of The Web Conference 2020_. 2020.

---

### Author Response · Authors · 2025-12-01
**Global Response**

Dear Reviewers and ACs,

We sincerely thank all reviewers for their thoughtful assessments and helpful suggestions. We are delighted that the reviewers acknowledge the technical contributions and performance advantages of our method. Specifically, reviewers acknowledged:

- **Clean and Modular Architecture.** The reviewers appreciated that our design removes rehearsal and avoids forgetting through a structured prototype hierarchy together with a lightweight hypernetwork. The division of roles, where task-prototypes for routing and class-prototypes with dynamic task-bias for finer separation, offers continual adaptation while avoiding storage of past data or excessive task-specific parameters. (Reviewer `X5bN`, Reviewer `YypD`, Reviewer `YdMv`)

- **Theory-Backed Knowledge Transfer.** Reviewers valued that we provide analysis explaining when the hypernetwork-generated heads can recover or closely approximate independently trained heads. The resulting risk transfer result offers insight into expected performance behavior during continual adaptation. (Reviewer `X5bN`, Reviewer `YdMv`)

- **Strong Forget-Free Performance.** Our method achieves zero average forgetting and sets a new performance level on all four GCIL benchmarks, while maintaining parameter efficiency. These findings are consistent across ablations and match the theoretical motivation. (Reviewer `YypD`, Reviewer `MttK`, Reviewer `YdMv`)

- **Clear and Well-Presented Work.** The pipeline is clearly explained in a straightforward manner, enabling the method to be easy to follow and reproduce. Training processes (prototype construction, dynamic bias, hypernetwork-based head generation) and inference with prototype-dictionary task identification are transparent and implementation-friendly. (Reviewer `X5bN`)

We have provided detailed responses to each reviewer’s comments and have addressed all concerns. Based on the reviewers’ suggestions, we have expanded our analysis of trainable parameter counts to better highlight the parameter-efficient nature of our method. We have also included additional experiments that evaluate robustness under different noise levels, verify accurate task-ID prediction, and demonstrate superior performance under larger numbers of tasks. Additional experiments, implementation details, and clarifications appear in the revised PDF (new results included in the appendix).

Once again, we would like to express our gratitude to all the reviewers for their valuable insights, which have greatly helped us improve our work.

---

### Meta-Review · Area_Chair_yVFV · 2026-01-24

**Summary:**

The paper presents a parameter-efficient framework for rehearsal-free graph class-incremental learning. The reviewers appreciated the modularity of the architecture as well as the theoretical results. However, there were essential concerns about the novelty and empirical advantages over TPP, the most closely related baseline. There were also concerns about the experimental scope being too narrow. The authors attempted to rebut these objections with additional experiments during the response period, but the reviewers don't seem swayed. Given this, I must recommend rejection this time around.

**Reviewer Concerns:**

See the original reviews.

**Reviewer Scores:**

The concerns about the experimental setup and novelty over TPP seem quite fundamental. The authors' additional data might have led to some improvement in the scores, but it is hard to imagine all the objections going away.

---

### Decision · Program_Chairs · 2026-01-26

Reject